# A nanounit strategy reverses immune suppression of exosomal PD-L1 and is associated with enhanced ferroptosis

Guohao Wang[1,2,4], Lisi Xie[1,2,4], Bei Li[1,2,4], Wei Sang[1,2], Jie Yan[1,2], Jie Li[1,2], Hao Tian[1,2], Wenxi Li[1,2], Zhan Zhang[1,2], Ye Tian[1,2] & Yunlu Dai ⬤ [1,2,3✉]

In addition to increasing the expression of programmed death-ligand 1 (PD-L1), tumor cells can also secrete exosomal PD-L1 to suppress T cell activity. Emerging evidence has revealed that exosomal PD-L1 resists immune checkpoint blockade, and may contribute to resistance to therapy. In this scenario, suppressing the secretion of tumor-derived exosomes may aid therapy. Here, we develop an assembly of exosome inhibitor (GW4869) and ferroptosis inducer ($Fe^{3+}$) via amphiphilic hyaluronic acid. Cooperation between the two active components in the constructed nanounit induces an anti-tumor immunoresponse to B16F10 melanoma cells and stimulates cytotoxic T lymphocytes and immunological memory. The nanounit enhances the response to PD-L1 checkpoint blockade and may represent a therapeutic strategy for enhancing the response to this therapy.

[1] Cancer Centre, Faculty of Health Sciences, University of Macau, Macau SAR 999078, China. [2] Institute of Translational Medicine, Faculty of Health Sciences, University of Macau, Macau SAR 999078, China. [3] MoE Frontiers Science Center for Precision Oncology, University of Macau, Macau SAR 999078, China. [4]These authors contributed equally: Guohao Wang, Lisi Xie, and Bei Li. ✉email: yldai@um.edu.mo

Tumor cells can evoke systemic immune perturbations for rapid expansion. Recent studies have discovered that beyond upregulating the expression of programmed death-ligand 1 (PD-L1) on cellular surface[1,2], tumor cells (e.g., melanoma) secrete a high level of PD-L1 on exosome, a particular form of extracellular vesicle derived from the cellular endocytic pathway[3], to interfere in systemic immune state. Exosomal PD-L1 can travel to the draining lymph node and inhibit T cells activity in an immune-suppressive modality[4]. Leaning in, this preemptive strategy may explain the resistance of exosomal PD-L1 to PD-L1/PD-1 immune checkpoint blockade therapy[5,6], and associate with a large percentage of relevant clinical failures[7]. The tactic to downregulate the secretion of these exosomes alleviates the exhaustion in lymph nodes through restoring and proliferating functional T cells[4], which elevates the anti-tumor efficiency of immunotherapy.

Ferroptosis is generally accompanied by a rich accumulation of intracellular iron and generation of oxidative hydroxyl radicals, impairing the antioxidant capacity in cells for lipid peroxidation[8,9]. This oxidative cell death has been evidenced to involve in a variety of therapeutic scenarios, especially T cell immunity and cancer immunotherapy[10]. For instance, ferrototic cancer cells have been discovered to release high mobility group box 1 (HMGB1) to modulate the immune-relevant inflammatory response[11] and to enable the activation and maturation of bone marrow-derived dendritic cells (BMDCs)[12]. Mutually, immune-activated T cells can release a high level of interferon-γ (IFN-γ) to intensify ferroptosis-specific lipid peroxides in tumor cells; and enhanced ferroptosis contributes to the anti-tumor immune efficacy. Inspired by the foregoing immunosupportive cases, we rationalized that directly linking the immunogenic superiority of exosome inhibition and ferroptosis may establish a potent immunotherapeutic strategy. Up to now, no study focuses on this linkage. The proposed approach is suspected to diminish the systemic immunosuppression caused by exosomal PD-L1, magnify the ferroptosis of tumor cells and upgrade the systemic anti-tumor immune response.

In this work, small molecule GW4869 is applied for exosome inhibition[13,14]. The hydrophobic nature of this exosome inhibitor hints us to conjugate hydrophilic hyaluronic acid (HA) with hydrophobic 5β-cholanic acid (CA), functionalizing this synthesized HACA carrier to be amphiphilic firstly (Fig. 1a). $Fe^{3+}$, as our ferroptosis inducer, is coordinated onto the HACA by polyphenol, forming HACA-Fe nanoparticles (NPs). Encapsulating GW4869 exosome inhibitor into the hydrophobic phase of HACA-Fe achieves our final product of HGF nanounit. As illustrated in Fig. 1b, inhibition performance of GW4869 decreases the secretion of tumor-derived exosome and weakens the firepower of exosomal PD-L1 indirectly. Immune-active T cells are thus shielded well to secret reactive IFN-γ cytokine, reducing cancer cellular SLC7A11 and SLC3A2, cystine/glutamate transporters. They both show pivotal roles in maintaining the cellular uptake to cystine and glutamate for anti-oxidation per se, whose downregulation inevitably enhances lipid peroxidation. Moreover, enriched cellular iron contributed by our HGF strengthens the cancer cellular ferroptosis further. To examine the anti-tumor performance of HGF, multiple B16F10 melanoma models are established. Individual HGF inhibits tumor growth and stimulates long-lasting immunological memory successfully. Combination with PD-L1 checkpoint blockade, HGF remedies the therapeutic limitations of free antibodies, including functional optimization of T cells, and suppression of tumor metastasis. The HGF designed is heralded as an attractive candidate for next-generation cancer immunotherapy.

## Results

**Synthesis and characterization of HGF.** Amphiphilic HACA was self-assembled in an aqueous solution[15] to be a yarn-like hydrophilic nanoparticle with a hydrophobic core, whose further functionalization with polyphenol-iron achieved HACA-Fe. GW4869 inhibitor can be further encapsulated into the hydrophobic inner core of HACA-Fe (HGF) proportionately (1:9, 1:4, and 2:3, w/w, Supplementary Fig. 1 and Table 1) via a simple ultrasonication. Finally, nanospherical HGF (Fig. 2a) with the highest GW4869 loading efficiency (81.8 ± 3.3%) was applied in the following studies with 20.9 wt % of Fe. Elemental iron, carbon, and oxygen were distributed universally (Fig. 2b, c) in HGF NPs with a mean diameter of 104.5 ± 7.7 nm (Fig. 2d). Constructed HGF NPs exhibited a variation of zeta potential from – 33.4 mV (HACA) to – 17.0 mV (Fig. 2e), emphasizing the iron intercalation, and showed a characteristic UV peak of GW4869 at ~ 360 nm wavelength, indicating the successful encapsulation of GW4869 (Fig. 2f). HGF NPs can be decomposed by intracellular hyaluronidase. Here, incubating HGF NPs with hyaluronidase-1 (Hyal) expanded these nanounits into microcomplexes within 20 min (Fig. 2g), which indicated the Hyal-triggered collapse of the HA backbone. A burst release of 80% of GW4869 within 6 h was thus detected (Fig. 2h). Released GW4869 had no obvious cytotoxicity to B16F10 cancer cells in vitro (Supplementary Fig. 2). Differently, a certain level of cell death was induced by iron composite from HGF NPs (34.5% of B16F10 cell death at 100 μM for 48 h), which might be attributed to a $Fe^{3+}$-relevant ferroptosis. Then we detected the direct effect of drugs on CD8+ T cells in vitro (Supplementary Fig. 3a, b). At the highest concentration of HGF NPs, the IFN-γ production and vitality of CD8+ T cells were kept well (vitality, 81.2% at 100 μM for 48 h).

As HA can bind to a cluster of differentiation protein 44 (CD44), a receptor that is overexpressed on various tumor cells[15,16], HA-based HGF NPs can target tumor cells with CD44 overexpression[17] very well, which ascertain that GW4869 functions in tumor area mainly. HGF NPs targeted B16F10 cancer cells with CD44 overexpression[18,19] in Fig. 2i, which was evidenced via an uptake inhibition induced by the pre-blocking of free HA. Effective intratumoral penetration of HGF NPs was thus anticipated. Here, an in vitro three-dimensional multicellular spheroid model (MCS) was built to co-culture with ICG labeled HGF NPs for 24 h, a universal fluorescent spheroid was acquired. Specifically, apart from top localization, HGF NPs accumulated in the MCS core (Supplementary Fig. 4), scanned by z-stack confocal laser scanning microscopy (CLSM) from MCS top to bottom. Deep penetration of HGF NPs to solid cellular spheroid was exhibited in these z-axis images. Furthermore, in vivo tumor accumulation of HGF NPs was evaluated in C57BL/6 J female mice bearing subcutaneous B16F10 tumor model. Fluorescence imaging in Fig. 2j and Supplementary Fig. 5 indicated that ICG labeled HGF NPs enriched in tumor region rapidly and grew to a plateau at 12 h post intravenous (i.v.) administration. Quantitatively, 15.7% of intratumoral HGF NPs was found at the maximum (Fig. 2k). In contrast to HGF NPs, the tumoral fluorescence signal in free ICG group was barely noticed. Detailed pharmacokinetics of HGF NPs was assessed as well. The half-lives of biodistribution and elimination of HGF NPs were 2.97 h and 62.79 h, respectively (Supplementary Fig. 6), indicating the long blood circulation of HGF NPs.

**Inhibition of exosomal PD-L1 and immunostimulation by HGF.** As shown in the Supplementary Fig. 7, the expression level of PD-L1 on exosome consisted with that on the surface of B16F10 cells. Western blot was applied to evaluate the influence of GW4869 to exosome production in vitro. As shown in Fig. 3a

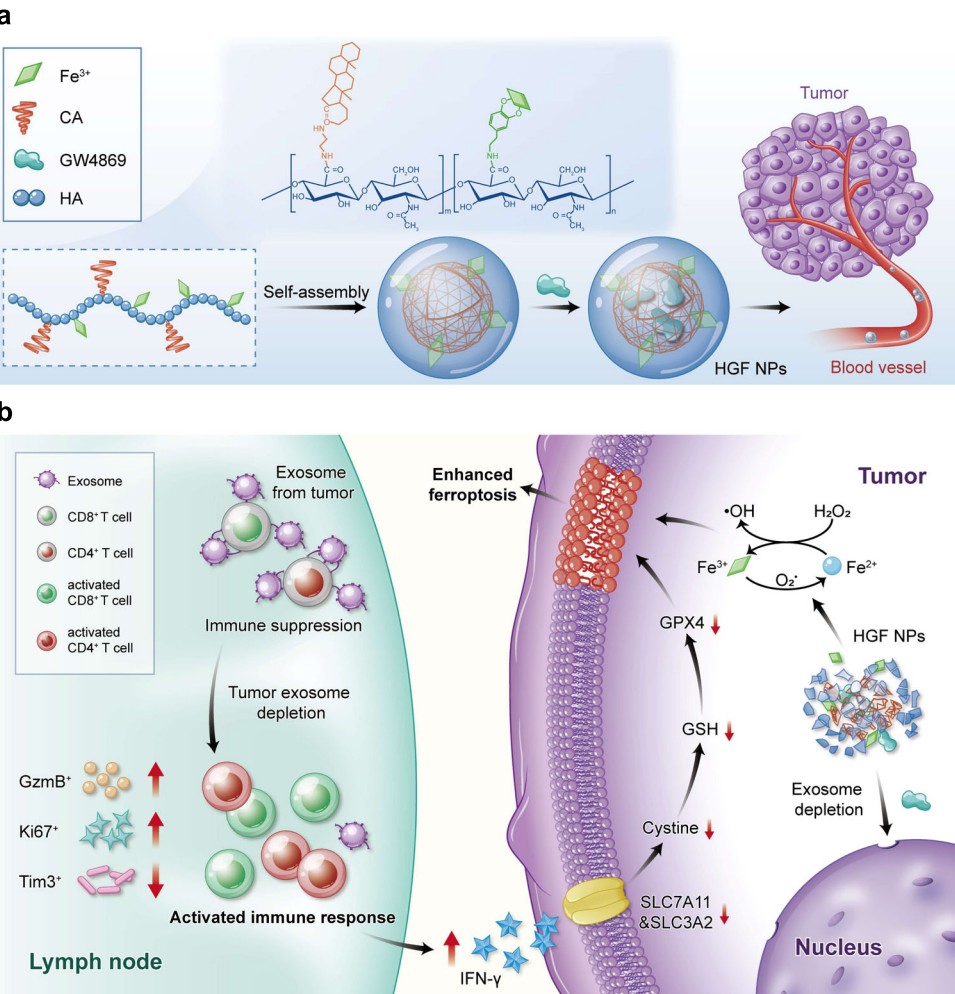

**Fig. 1 Schematic illustration of HGF-relevant preparation and therapeutic strategy. a** The exosome inhibitor (GW4869) with ferroptosis inducer (iron ion) were unified into a HA (hyaluronic acid) based nanoplatform (HGF NPs). **b** With intravenous administration released GW4869 from HGF NPs significantly decreased tumor-derived exosome generation, resulting in activated anti-tumor immune response and a robust memory response against tumor cells. Furthermore, re-activated T cells released a high level of IFN-γ (interferon-γ) cytokine to downregulate the expression of SLC7A11 and SLC3A2, both of which are responsible for the synthesis of intracellular anti-oxidant cystine, enhancing the ferroptosis induced by an enriched iron source from HGF NPs.

and Supplementary Fig. 8, HACA-Fe-based ferroptosis had a moderate effect on the reduction of exosomes. GW4869 from HGF NPs diminished the exosome biomarker CD63[20] of B16F10 cells significantly, much more effective than the function of free GW4869. A decrease in exosome secretion reduced the exosomal PD-L1 dramatically. Next, B16F10 tumor-bearing mice were used to determine the exosome inhibition of HGF NPs in vivo. After three treatments, tumors in all groups were collected at day 7 (Fig. 3b). By isolating exosomes from tumor tissues, we noticed that the CD63 and the PD-L1 in the HGF NPs group was barely visible (Fig. 3c), suggesting the remarkable inhibition of GW4869 to the tumoral exosome and exosomal PD-L1.

Significant elimination of tumoral exosome may rock the systemic anti-tumor immune response. Tumor-draining lymph nodes (TDLNs) were thus collected on day 7 for analysis. As expected, striking differences were concluded among the five treatment groups (PBS, HACA-Fe, HACA-GW, HGF, and HGF plus liproxstatin). Briefly, HACA-GW NPs promoted 36.7% of DC maturation (Supplementary Figs. 9 and 10). Integrating with iron enhanced DC maturation to the highest level (43.9%, in HGF group), which might be explained by the Fe³⁺-induced ferroptosis. The addition of ferroptosis inhibitor liproxstatin

downregulated the DC maturation again, supporting the assistance of ferroptosis to immune stimulation. Correspondingly, similar variations were found in T-cell stimulation. CD8⁺ and CD4⁺ T cells of HGF NPs group occupied the greatest fractions of T cells in all treated groups (Fig. 3d–f and Supplementary Fig. 11). Their activation, proliferation, and even exhaustion were further analyzed statistically. In particular, activation marker Granzyme B (Fig. 3g–i and Supplementary Figs. 11 and 12) and proliferation marker Ki67 (Fig. 3j, k and Supplementary Figs. 13 and 14) showed the strongest intensities in TDLNs treated by HGF NPs. Apart from enriching the functional T cells, HGF NPs attenuated their exhaustion (Fig. 3l, m and Supplementary Fig. 15) concurrently, which was demonstrated by lowest Tim3 (exhaustion marker) degree. Altered functional cell population may affect the systemic cytokine profiles as well[21]. Herein, sera were collected to delineate the variation curves of pro-inflammatory cytokines (tumor necrosis factor-α (TNF-α), interleukin-6 (IL-6), and interleukin-12 (IL-12)). Consistent with immunostimulation in TDLNs, mice treated with HGF NPs secreted a considerable amount of TNF-α, IL-6, and IL-12 cytokines (Supplementary Fig. 16). Together, unifying tumoral exosome inhibition and ferroptosis showed high

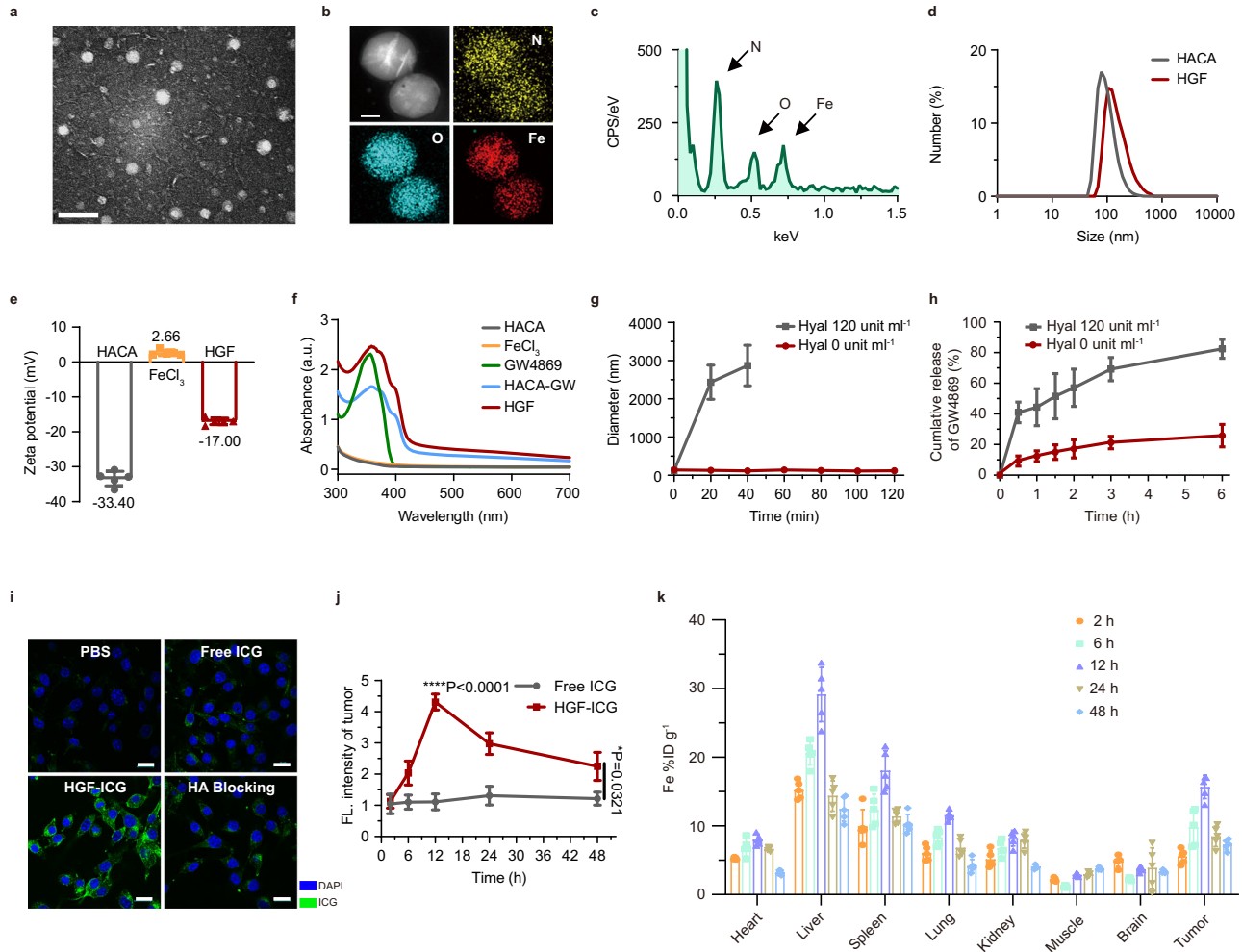

**Fig. 2 Characterization and biodistribution of HGF NPs. a–c** Transmission electron microscopy (TEM) image (**a** scale bar: 200 nm), representative elemental mapping (**b** scale bar: 20 nm) and energy-dispersive X-ray spectroscopy (EDS) analysis **c** of HGF NPs. Images were representative of three experiments. **d** The size of HACA and HGF NPs determined by dynamic light scattering (DLS). Experiments were performed five times independently with similar results. **e** Zeta potential of HACA, FeCl₃, and HGF NPs. $n = 5$ biologically independent samples per group at each time point. Data were presented as mean ± SD. **f** The UV–Vis absorption spectra of HACA, FeCl₃, GW4869, HACA-GW, and HGF NPs. Experiments were performed three times independently with similar results. **g** and **h** Particle size changes **g** of HGF NPs and release patterns **h** of GW4869 from HGF NPs in the presence or absence of Hyal (hyaluronidase-1). $n = 3$ biologically independent samples per group at each time point. Data were presented as mean ± SD. **i** Cell uptake of HGF NPs labeling with ICG. Scale bar: 20 μm. Images were representative of three experiments. **j** Quantification of the corresponding fluorescence intensity of tumor sites at 2, 6, 12, 24, and 48 h after injection with free ICG and ICG labeled HGF NPs. $n = 4$ biologically independent animals per group at each time point. Data were presented as mean ± SD. Statistically significant differences between groups were identified by one-way ANOVA with Tukey's post hoc test. $*P < 0.05$, $**P < 0.01$, $***P < 0.001$, and $****P < 0.0001$. **k** Biodistribution of HGF NPs in various tissues after 2, 6, 12, 24, and 48 h intravenous administration. HGF NPs concentrations were normalized as the percentage of the injected dose of Fe per gram of each organ (%ID g⁻¹). $n = 5$ biologically independent animals per group at each time point. Data were presented as mean ± SD.

potential to trigger strong anti-tumor immune responses and therapeutic efficacy.

**HGF- enhanced tumor cellular ferroptosis**. Further analyzing immune tests in Fig. 3, we found that combining ferroptosis inhibitor liproxstatin with HGF NPs reduced the functional T fractions apparently. Ferroptosis was thus revealed to enhance T cell activation and proliferation. It has been reported that immune-activated T effectors can release IFN-γ and elevate ferroptosis-specific lipid peroxidation[10]. Moreover, the exosomal PD-L1 can inhibit the cytokine IFN-γ production of CD8+ T cells since this inhibition can be nearly abolished by the pre-treatment of exosomes with anti-PD-L1 antibodies (Supplementary Fig. 17). In this context, we wonder that whether inhibiting the exosomal PD-L1, an immune-active strategy demonstrated from our

foregoing tests can influence cancer cellular ferroptosis concurrently. IFN-γ level was firstly investigated via stimulating CD8+ T cells with the exosomes collected from the B16F10 medium co-culturing with PBS, GW4869, or HGF NPs (Fig. 4a). A significant increase in IFN-γ production was detected while augmenting HGF concentration (Fig. 4b and Supplementary Fig. 18). Through collecting IFN-γ-containing cultural medium to treat B16F10 cells, different intracellular lipid ROS was measured (Fig. 4c). A magnificent increase of lipid ROS level occurred in the HACA-Fe group, which was attributed to the Fe³⁺-induced ferroptosis. Integrating GW4869 into the HACA-Fe system enhanced the lipid ROS further, which may elicit robust ferroptosis in HGF group. To verify this hypothesis, several inhibitors were adopted to distinct the cell death modalities. As shown in Supplementary Fig. 19, ferroptosis inhibitor (ferrostatin-1, Fer-1)

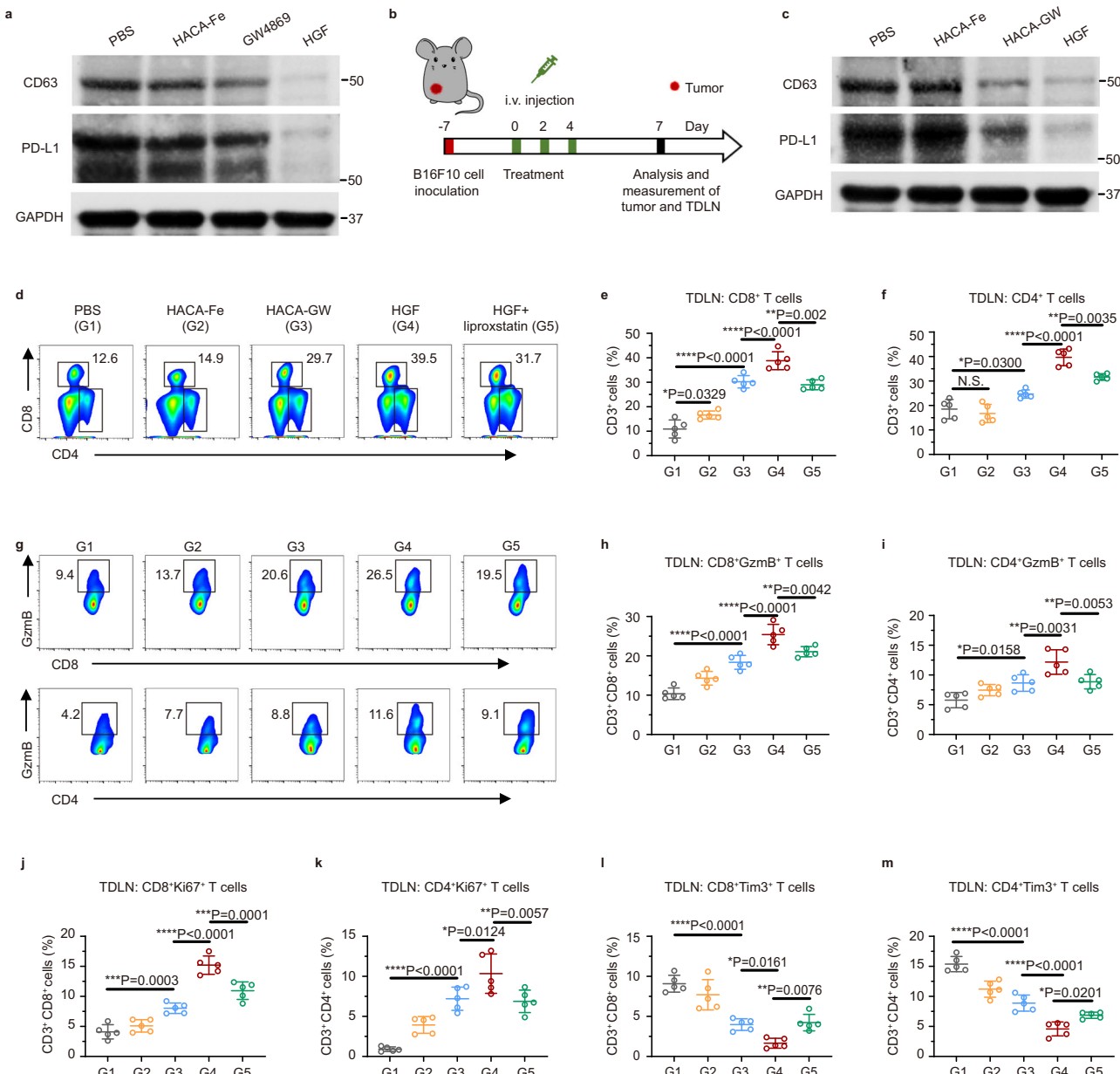

**Fig. 3 HGF NPs inhibited exosomal PD-L1 and triggered robust immunostimulation. a** Western blot analysis of exosome marker CD63 and PD-L1 via collecting the exosomes from the cell-free media of B16F10 cells with various treatments. GAPDH served as a loading control. Images were representative of the three experiments. **b** Scheme of in vivo experimental design for (**c**–**m**). C57BL/6 mice were subcutaneously inoculated with $1 \times 10^6$ B16F10 cells. When tumor volumes reached ~100 mm³, mice were treated with PBS, HACA-Fe, HACA-GW, or HGF on day 0, day 2, and day 4. Tumor and tumor-draining lymph node (TDLN) were collected on day 7 for further analysis. **c** Western blot analysis of exosome marker CD63 and PD-L1 in tumor tissues after treatment. Images were representative of the three experiments. **d**–**f** Flow cytometric plots (**d**) and quantification of CD8+ (**e**) and CD4+ (**f**) T cells, respectively, gated by CD3+ T cells in the TDLN ($n = 5$) in groups of PBS (G1), HACA-Fe (G2), HACA-GW (G3), HGF (G4), and HGF + liproxstatin (G5). **g**–**m** Flow cytometric analysis of granzyme B+ (GzmB+, **g**–**i**), Ki67+ (**j** and **k**), and Tim3+ (**l** and **m**) in CD8+ and CD4+ T cells. $n = 5$ biologically independent animals per group. Data were presented as mean ± SD. Statistically significant differences between groups were identified by one-way ANOVA with Tukey's post hoc test. *$P < 0.05$, **$P < 0.01$, ***$P < 0.001$, and ****$P < 0.0001$.

restrained the B16F10 cancer-killing capacity of HGF remarkably, whereas the apoptosis inhibitor (carbobenzoxy-valyl-alanyl-aspartyl-[O-methyl]-fluoromethylketone, Z-VAD-FMK) and the necroptosis inhibitor (necrostatin-1s, Nec) did not. In a short summary, the greatest IFN-γ secretion induced the highest lipid ROS, which elicited the strongest ferroptosis in the HGF group. Lipid peroxidation associates with the level of intracellular glutathione[22–24], whose synthesis is mediated by SLC7A11 and SLC3A2, cystine/glutamate transporter proteins[5,25]. In Fig. 4d, visible decreasing of SLC7A11 and SLC3A2 was shown in the

HGF group assisted by IFN-γ. GPX4, which plays a key role in lipid repair systems, can be inactivated by cystine/glutamate depletion, thereby inducing the ferroptosis[8]. As shown in Fig. 4e, the activity of GPX4 in HGF NPs decreased substantially, compared with other groups. These tests revealed that IFN-γ impeded cellular anti-oxidative protection and accelerated lipid peroxidation.

Next, a B16F10 melanoma tumor-bearing mouse model was established (Fig. 4f) to evaluate the ferroptosis-related signals in vivo. Mice were treated with PBS, HACA-Fe, HACA-GW, and

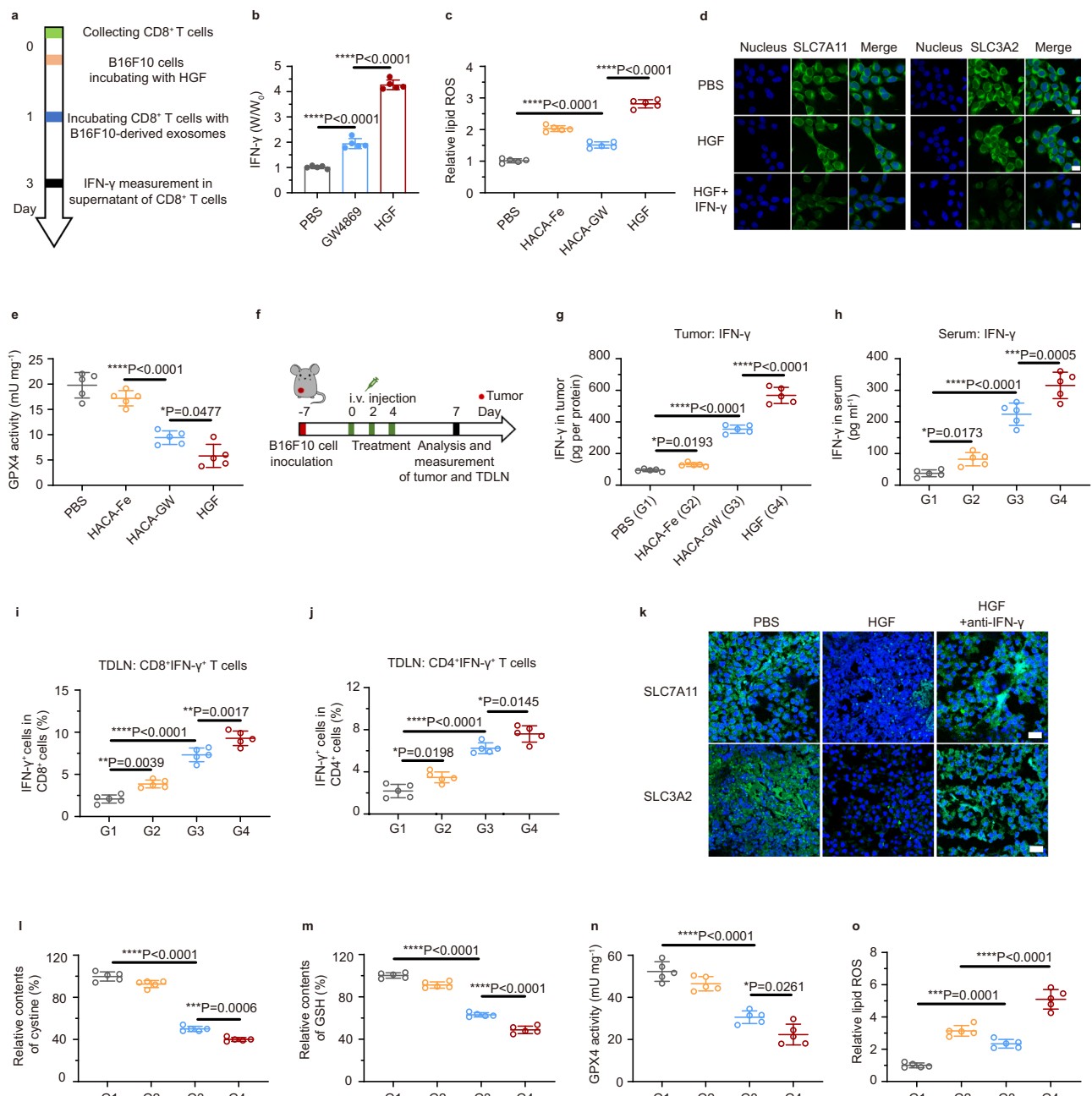

**Fig. 4 Evaluation of enhanced ferroptosis induced by HGF-relative exosomal PD-L1 inhibition. a** In vitro schedule for ferroptosis-relevant tests. **b** Relative IFN-γ release from CD8[+] T cells incubated in different conditions. $n = 5$ biologically independent samples per group. **c** Relative lipid ROS of B16F10 cells co-cultured with CD8[+] T cells after incubating in PBS, HACA-Fe, HACA-GW, and HGF NPs for 48 h. $n = 5$ biologically independent samples per group. **d** CLSM observation of expression of SLC7A11 and SLC3A2 in B16F10 cells treated with PBS, HGF or HGF + IFN-γ for 24 h. Scale bar: 20 μm. Experiments were performed three times independently with similar results. **e** GPX4 activity in B16F10 cells co-cultured with CD8[+] T cells after incubating in PBS, HACA-Fe, HACA-GW, and HGF NPs. $n = 5$ biologically independent samples per group. **f** Schematic showing the experiment design for in vivo evaluations (**g–o**). $n = 5$ biologically independent animals per group. **g** The intratumoral IFN-γ level after treatment. G1, PBS; G2, HACA-Fe; G3, HACA-GW; G4, HGF. **h** IFN-γ secretion in serum. **i** and **j** Flow cytometric quantification of IFN-γ[+] T cells after gating on CD3[+]CD8[+] T cells (**i**) and CD3[+]CD4[+] T cells (**j**) in TDLN. **k** Immunofluorescence assay for the expression of SLC7A11 and SLC3A2 in tumor tissues. Scale bar: 50 μm. **l–o** Cystine (**l**), GSH (**m**), GPX4 activation (**n**), and lipid ROS (**o**) levels in B16F10 tumor tissue after treatment. All data are presented as mean ± SD. Statistically significant differences between groups were identified by one-way ANOVA with Tukey's post hoc test. *$P < 0.05$, **$P < 0.01$, ***$P < 0.001$, and ****$P < 0.0001$.

HGF, respectively. Consistent with in vitro cellular results, tumor tissues, and sera collected in HGF group contained highest amounts of IFN-γ (Fig. 4g, h). This cytokine is mainly secreted from CD8[+]IFN-γ[+] and CD4[+]IFN-γ[+] T cells[26,27], both of which occupied the largest ratios in the TDLNs from HGF group (Fig. 4i, j and Supplementary Figs. 20 and 21). By digging the

microscopic variations of HGF-treated tumors, cellular levels of SLC3A2, and SLC7A11 were reduced significantly (Fig. 4k), which obstructed the cellular intake of cystine (Fig. 4l) and the downstream synthesis of GSH (Fig. 4m); ultimately, lack of anti-oxidative GSH triggered the inactivation of the GPX4 (Fig. 4n) and finally severe lipid peroxidation (Fig. 4o). Treating tumors

with the combination of HGF and anti-IFN-γ agents restored the expression of SLC3A2 and SLC7A11, which suggested that the positive role of IFN-γ in cancer cellular ferroptosis. Herein, it should be noted that HGF NPs can enhance the GPX4 activity of T cells significantly, which may associate with the activation of $CD8^+$ and $CD4^+$ T cells (Supplementary Fig. 22).

**Anti-tumor performance of HGF in vivo.** To validate the therapeutic effect of our HGF nanounits in vivo, B16F10 tumor-bearing mouse model was applied here (Fig. 5a). Therapy was conducted when tumor volume reached circa 100 mm³. As is exhibited in Fig. 5b, c and Supplementary Figs. 23–26, weak ferroptosis of HACA-Fe hardly affected the tumor development; however, arming this function with tumor exosome elimination improved the therapeutic result remarkably, which was evidenced by the apparent difference of tumor inhibition induced by HACA-GW and HGF, respectively. And the HGF NPs triggered a longer survival period than the other treatments tested (Fig. 5d). Therapeutic biosafety can be reflected from the healthy organ status (Supplementary Fig. 27). In addition, the residual tumors were collected and the immunocellular compositions were analyzed. Notably, HGF NPs treatment induced the highest increase in tumor-infiltrating cytotoxic and helper T lymphocytes ($CD3^+CD8^+$ and $CD3^+CD4^+$) due to the combination of relieved immunosuppression of exosomal PD-L1 and enhanced ferroptosis

(Fig. 5e, f and Supplementary Fig. 28). To analyze the immunological memory after therapeutic treatment, spleens in all mice groups were collected. Mice in HGF group exhibited a 1.6-fold increase of memory T cell ($CD3^+CD8^+CD44^+CD62L^-$) populations compared with the PBS control, much better than the single function of HACA-Fe or HACA-GW (Fig. 5g, h and Supplementary Fig. 29).

**Therapeutic impetus of HGF to checkpoint inhibitor therapy.** Since PD-L1 checkpoint inhibitor therapy cannot hinder the immunosuppression of exosomal PD-L1 effectively[28], the therapeutic synergy of HGF and anti-PD-L1 antibodies was evaluated following the procedure in Fig. 6a. Treatments were conducted on day 7 when the tumor volume reached ~ 100 mm³. Free anti-PD-L1 antibodies exhibited limited inhibition to the tumor growth; however, its combination with HGF reversed the infaust situation completely (Fig. 6b, c). In addition, different from single PD-L1 checkpoint blockade therapy and HGF treatment, the combination of both of which further increased survival time (Fig. 6d). Apoptotic tumor cells in combination group were observed (Supplementary Fig. 30). To understand the underlying antitumor mechanism further, immune cells in the lymph node were collected. We found that the assistance of HGF NPs promoted DC maturation to a high level (Supplementary Fig. 31). In detail, HGF plus anti-PD-L1 matured 43.2% of DCs. In comparison, only 28.8% of DC maturation was detected in free antibody

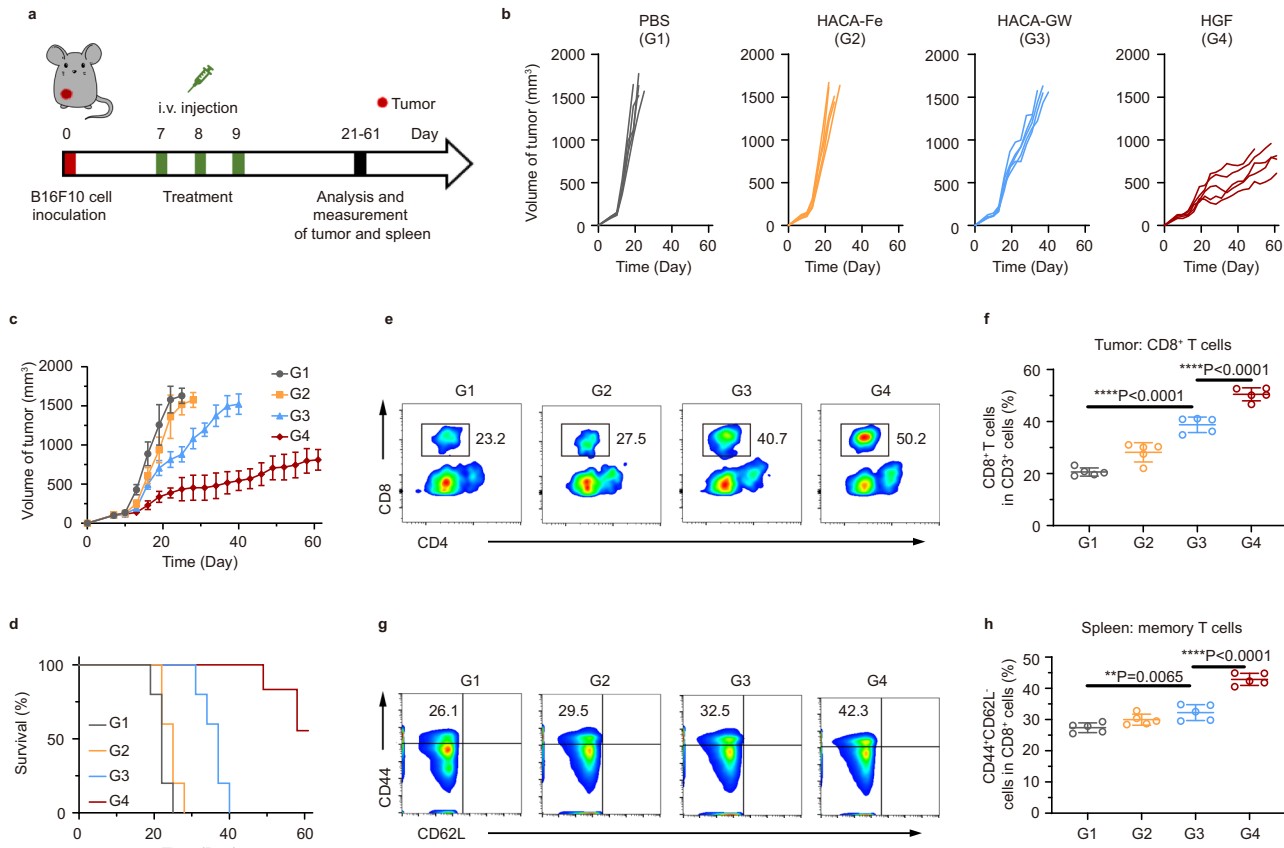

**Fig. 5 HGF induced an effective therapeutic effect against B16F10 tumor. a** Schematic illustration of the experiment design to assess the tumor inhibition and the immune responses as triggered by PBS, HACA-Fe, HACA-GW, or HGF NPs on subcutaneous B16F10 tumor model. G1, PBS; G2, HACA-Fe; G3, HACA-GW; G4, HGF. **b** and **c** Tumor growth curves during treatment. **d** Survival rates of different groups over time. **e** and **f** Representative flow cytometry plots and quantification of **e** CD8+ **f** and CD4+ T cells after gating on $CD45^+CD3^+$ T cells in tumor tissue. **g** and **h** Representative flow cytometric analysis of memory T cells ($CD3^+CD8^+CD44^+CD62L^-$, gated on $CD3^+CD8^+$ T cells) in the spleen. Data were presented as mean ± SD. $n = 5$ biologically independent animals per group. Statistically significant differences between groups were identified by one-way ANOVA with Tukey's post hoc test. *$P < 0.05$, **$P < 0.01$, ***$P < 0.001$ and ****$P < 0.0001$.

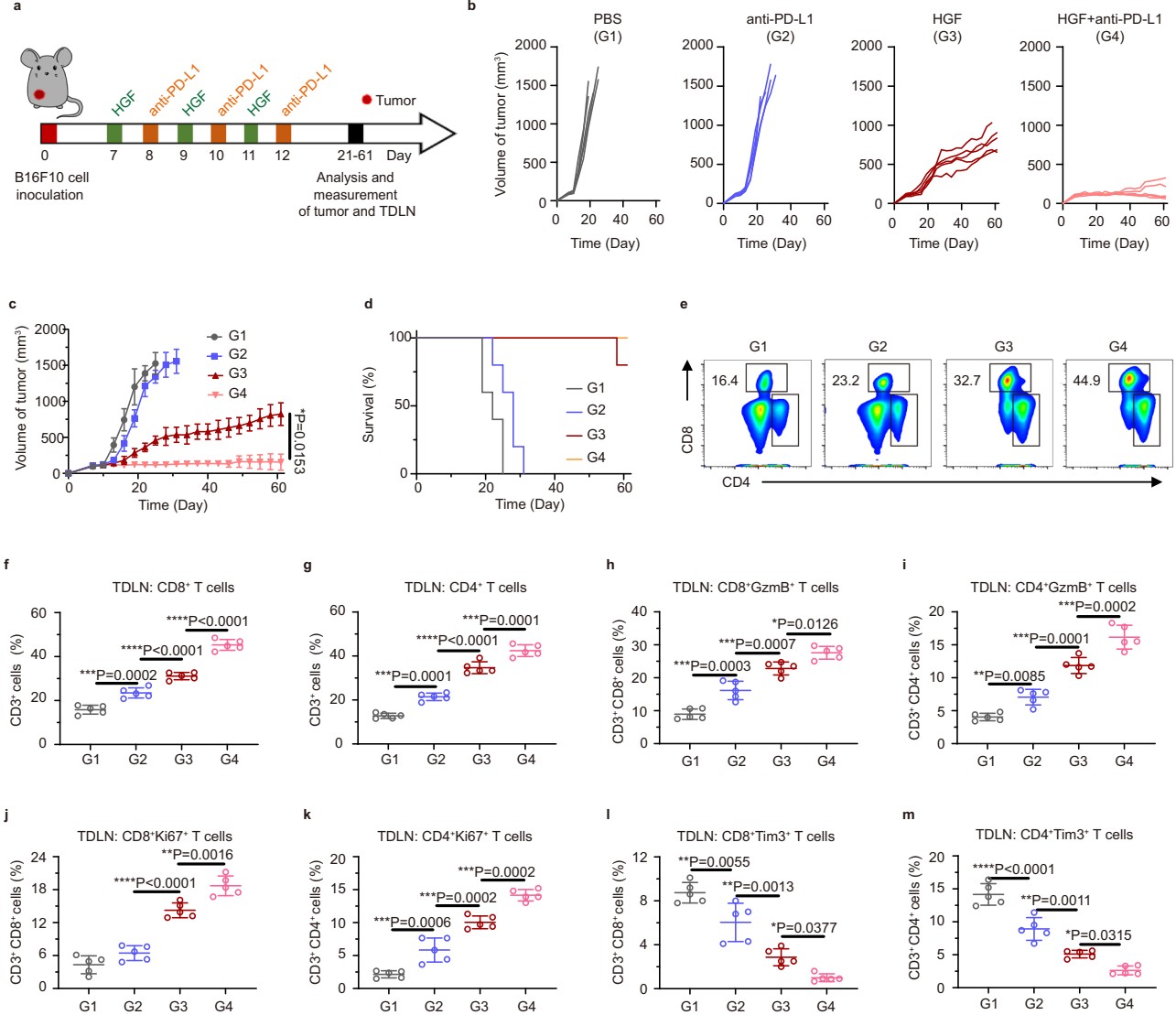

**Fig. 6 In vivo therapeutic impetus of HGF to PD-L1 checkpoint inhibitor therapy. a** Schematic illustration of the experiment design to assess the tumor inhibition and the immune responses as triggered by PBS, anti-PD-L1, HGF or HGF + anti-PD-L1 on subcutaneous B16F10 tumor model. G1, PBS; G2, anti-PD-L1; G3, HGF; G4, HGF + anti-PD-L1. **b** and **c** Tumor growth curves during treatment. **d** Survival rates of different groups over time. **e**–**g** Flow cytometric plots and quantification of **e** CD8$^+$ and **f** CD3$^+$ **g** T cells, among CD3$^+$ T cells in the TDLN. **h**–**m** Flow cytometric quantification of GzmB$^+$ cells (**h**–**i**), Ki67$^+$ cells (**j** and **k**) and Tim3$^+$ cells (**l** and **m**) in CD8$^+$ and CD4$^+$ T cells. Data were presented as mean ± SD. $n = 5$ biologically independent animals per group. Statistically significant differences between groups were identified by one-way ANOVA with Tukey's post hoc test. *$P < 0.05$, **$P < 0.01$, ***$P < 0.001$, and ****$P < 0.0001$.

group. Next, downstream T cell stimulation showed statistical differences. As shown in Fig. 6e–g, the absolute percentage of CD8$^+$ cells in the combined treatment occupied 45.3%, an ~ 2-fold improvement in comparison with that of anti-PD-L1 group (23.5%). A higher CD4$^+$ cells ratio was also detected in the TDLNs of HGF plus anti-PD-L1 group. The apparent activation, proliferation, and low exhaustion of T cells were further explored by analyzing the cellular activation marker Gramzyme B, proliferation marker Ki67, and exhaustion marker Tim3 (Fig. 6h–m and Supplementary Figs. 32–36).

To explore the anti-tumor immunological memory after therapy, spleens in all groups were examined by flow cytometric analysis. Different from the free antibody group, a 1.5-fold population increasing of memory T cells (CD3$^+$CD8$^+$CD44$^+$CD62L$^-$) was stimulated in the combined group (Supplementary Fig. 37). Even so, low inflammatory spleens (Supplementary Fig. 38) can be observed in this optimal group.

In this in vivo trial, another intriguing phenomenon was the tremendous variation of tumor metastasis in the nearest draining lymph node from each group (Supplementary Fig. 39). Obviously, black melanoma cells infiltrated the lymph nodes in control and free antibody groups severely. On the contrary, no black tumor cells can be observed visually. Therefore, apart from the successful anti-tumor immune stimulation, HGF may inhibit the systemic metastasis of hypermetastatic B16F10 cancer effectively.

**Anti-metastatic enhancement of HGF plus PD-L1 blockade.** To ascertain the anti-metastatic capacity of HGF we designed, a lung-metastatic B16F10 model was established by injecting B16F10-FLuc cells into C57BL/6 mice intravenously (Fig. 7a). Bioluminescence imaging was adopted to trace the body distribution of these malignant tumor cells. Fearful lung accumulation of B16F10 cancer cells was represented in Fig. 7b, c. From

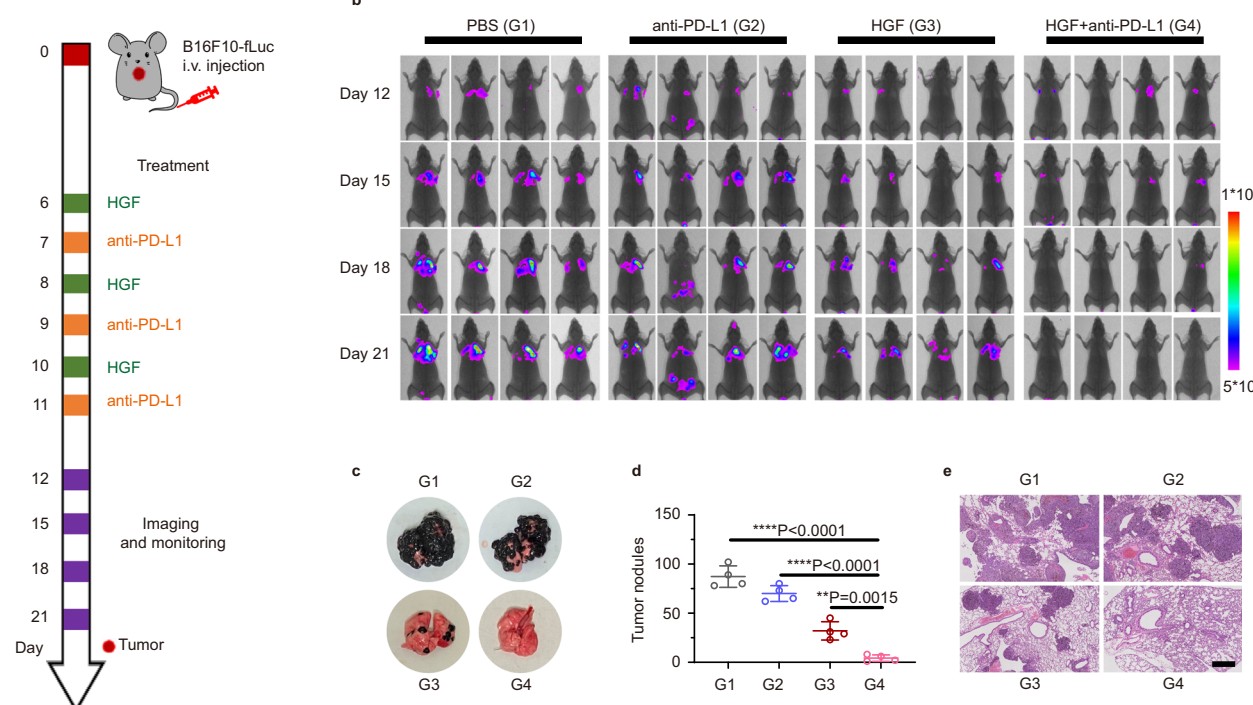

**Fig. 7 Combined HGF and anti-PD-L1 immunotherapy inhibited melanoma metastasis effectively. a** C57BL/6 mice were intravenously (i.v.) injected with $1 \times 10^5$ B16F10-FLuc cells, treated with PBS, anti-PD-L1, HGF (day 6, day 8, and day 10), or HGF every 2 days from day six for three times, and anti-PD-L1 on day 7, day 9, and day 11 (HGF + anti-PD-L1). **b** Respective bioluminescence images of metastatic B16F10-FLuc cells in mice body on day 12, day 15, day 18, and day 21. **c** Representative photographs of tumor nodules in lungs on day 21. **d** Number statistics of tumor nodules on lungs on day 21 after treatment. **e** H&E-stained slice images of lungs. Scale bar: 400 μm. Data were presented as mean ± SD. $n = 4$ biologically independent animals per group. Statistically significant differences between groups were identified by one-way ANOVA with Tukey's post hoc test. *$P < 0.05$, **$P < 0.01$, ***$P < 0.001$, and ****$P < 0.0001$.

the therapeutic trial, treating these tumor-bearing mice with free PD-L1 antibodies by three doses cannot slow the metastatic speed down, whereas HGF paused the tumor cellular diffusion rapidly. Unfortunately, the quick action of HGF only worked at the early stage of treatment, which was revealed by the tumor relapse from day 18. Here, combining HGF and anti-PD-L1 antibodies reduced the number of tumor nodules from 31 to 8 (Fig. 7d and Supplementary Fig. 40). By prolonging the feeding duration to 21st day, no bioluminescence signal was detected in this best treatment group (Supplementary Fig. 41). The lungs were further sectioned and stained with H&E for pathological analysis. As shown in Fig. 7e, lungs in the PBS and anti-PD-L1 groups were occupied by dense tumor nodules. Differently, these nodule populations were decreased by HGF to scattered level; a combination with PD-L1 blockade virtually eliminated the tumor metastasis in the lung area, indicating that the combined HGF/immune checkpoint blockade strategy activated a striking anti-lung metastasis effect to B16F10 melanoma.

## Discussion

Via studying the low patient response to PD-1/PD-L1 immune checkpoint blockade therapy, researchers have realized that exosomal PD-L1, secreted from metastatic melanomas, systematically suppresses the immune therapy[28–30]. Hereto, the approach to blockade exosomal PD-L1 genetically was harnessed to restore the anti-tumor immunity and lengthen the animal survival[4]. However, direct genetic engineering may bring unpredictable and irremediable consequences. In our study, standing nuclei gene unperturbed, exosome inhibitor GW4869 was adopted to downregulate the exosome secretion from B16F10 melanoma cells. Assembling this small molecule into an amphiphilic HA

nanoplatform diminished the exosomal PD-L1 remarkably. Systemic T cells were activated, therefore. To analyze the T cell differentiation in a tumor-draining lymph nodes, the fraction of cytotoxic T lymphocytes increased a lot; T activation marker Granzyme B, and proliferation marker Ki67 enriched; the population of memory T cells ($CD3^+CD8^+CD44^+CD62L^-$) grew strikingly. These variations suggested that inhibition of exosome secretion by GW4869 triggered systemic anti-tumor immunity and durable immune memory.

Apart from effective exosome inhibitor, ferroptosis inducer $Fe^{3+}$ was also unified into the HGF nanogroup, due to that this kind of oxidative cell death has been claimed to associate with T cell immunity and cancer immunotherapy. In our actual investigation, cross-fertilization between exosomal PD-L1 elimination and tumoral ferroptosis was revealed. Inhibiting the secretion of PD-L1-based exosome upregulated the IFN-γ level, which aided the downregulation of SLC3A2 and SLC7A11, two subunits to maintain the cysteine uptake by tumor cells. Ultimately, intracellular GSH content reduced significantly, leading to enhanced tumor cell lipid peroxidation/ferroptosis. On the contrary, inhibiting the HGF-induced ferroptosis with liproxstatin could suppress systemic immune activity to a low state. Our study sheds light on a previously unrecognized cooperative anti-tumor immune stimulation between the both, which is encouraged to explore in future work.

Until now, we have to admit that diminishing exosomal PD-L1 might do not affect the status of tumor cellular PD-L1, the latter of which still behaves potent inhibition to immunotherapy. Thus, a combination treatment of HGF and free anti-PD-L1 antibody was conducted to explore their anti-tumor synergy. Beyond expectation, this combo not only promoted the T cell activation

and proliferation to a much higher level compared to single HGF made but impeded the hypermetastatic performance of metastatic B16F10 melanoma in the whole-body system. Summarily, targeted anti-exosomal therapy collaborated with systemic anti-PD-L1 therapy synergized to invoke a robust systemic immune response against highly metastatic tumor.

## Methods

**Materials**. Sodium hyaluronic acid (HA, 234 kDa) was purchased from Lifecore Biomedical Company (MN, USA). 1-ethyl-3-(3-(dimethylamino)propyl)carbodiimide (EDC) and N-hydroxysuccinimide (NHS) were obtained from J&K Company (Beijing, China). Dimethyl Sulphoxide, 5β-Cholanic acid (CA), and tetra-butylammonium hydroxide (TBA) were purchased from Sigma-Aldrich. GW4869 was purchased from Adooq. ICG-Sulfo-Osu (ICG) was obtained from Dojindo Molecular Technologies. Liproxstatin-1 was purchased from Cayman Chemical. Recombinant mouse IFN-γ (485-MI) were purchased from R&D. BODIPY 581/591 C11 and anti-IFN-γ (XMG1.2) blocking antibodies were purchased from Thermo Fisher Scientific.

**Synthesis and characterization of HGF NPs**. Amphiphilic HA-5β-cholanic acid conjugates (HACA) were prepared by a two-step method: the first hydrophobicity of HA by tetrabutylammonium salt and a second aldol condensation with aminated 5β-cholanic acid, as previously described[18,19]. Then, HACA and dopamine hydrochloride (DA) conjugates (HACA-DA) were synthesized with 1-ethyl-3-(3′-dimethyl aminopropyl) carbodiimide (EDC) as an activation agent of carboxyl groups on the HA chain. Briefly, 1 g of HACA was dissolved in 100 mL of PBS solution and pH was adjusted to 8.0 using 1.0 M NaOH solution under a nitrogen atmosphere. After that, 575 mg of EDC and 345 mg of NHS were added. After 10 min, 1 g of DA was added without air bubbles after the pH of the reaction solution was adjusted to 5.5. Continuous dialysis (100 kDa) was conducted in ultrapure water to remove unreacted chemicals and byproducts. Next, at room temperature, via mixing iron ions and HACA-DA aqueous solution with overnight stirring and nitrogen protection, HACA-Fe NPs were obtained. At last, HGF NPs was readily prepared by a simple dispersion of GW4869 dissolved in Dimethyl Sulphoxide and HACA-Fe NPs in ultrapure water by high-intensity sonication on ice for 20 min. The final suspension was dialyzed in methanol/ultrapure water (1:1, v/v, 12 h) and ultrapure water (12 h) to remove free GW4869 and organic solvent, followed by lyophilization.

HGF morphology was imaged by Transmission electron microscopy (TEM), whose elemental composition was examined by element-mapping analysis. Particle size and zeta potential were analyzed using dynamic light scattering (DLS). The hyal-triggered degradation behaviors and GW4869-release profiles were investigated in an acetate buffer with different concentrations (0 or 120 unit ml$^{-1}$ of Hyal). HGF NPs were dispersed into cellulose ester dialysis tubes (molecular weight cutoff = 10,000 unit) and immersed in 30 ml of acetate buffer. Gentle shaking was followed at 100 rpm in 37 °C of water bath. The medium was collected time-dependently to measure the GW4869 releasing by UV spectrometry at 360 nm.

**Cell lines and culture media**. B16F10 cell lines were purchased from ATCC and maintained in Dulbecco's Modified Eagle Medium/Nutrient Mixture F-12 Medium supplemented with 10% FBS and 1% antibiotic solution. B16F10-FLuc cell lines were kindly provided by Dr. Lisi Xie at the University of Macau and maintained in Dulbecco's Modified Eagle Medium/Nutrient Mixture F-12 Medium supplemented with 10% FBS and 1% antibiotic solution. Mouse CD8$^+$ T cells were isolated from the spleen using EasySep Mouse CD8$^+$ T Cell Isolation Kit (Stemcell, Cat No. 19853). Then cells were maintained in Roswell Park Memorial Institute (RPMI) 1640 Medium plus 10% FBS and 1% antibiotic solution. Media and standard additives were purchased from Gibco® (Thermo Fisher Scientific). Cell lines were tested negative for mycoplasma by PCR analysis of cell culture supernatants. Cell cultures were maintained at 37 °C with 5% CO$_2$.

**Cell internalization and 3D penetration of HGF NPs**. B16F10 cells were cultured in 35 mm confocal dishes at $2 \times 10^5$ cells. In the next day, cells were washed by PBS for three times and incubated with free ICG or ICG labeled HGF (HGF-ICG) at 37 °C for 6 h with 5% of CO$_2$ atmosphere. For the blocking test, free HA was added to cells 40 min before HGF-ICG was applied. After incubation, all cells were washed with cold phosphate buffered saline (PBS) thoroughly. The cells were finally fixed in cold ethanol for 15 min and mounted with DAPI for 10 min. Finally, cell uptake of HGF was observed by a confocal laser scanning microscopy (CLSM, Nikon A1R) and images were analyzed by Zen software.

To evaluate the penetration of HGF NPs in vitro, a 3D multicellular B16F10 cells model was prepared. Firstly, B16F10 cells were seeded in Corning® spheroid microplates plates at 2000 per well for 5 days to yield B16F10 cancer cellular spheroids. The culture medium was replaced by HGF-ICG and co-cultured for another 24 h. And then, 4 wt % paraformaldehyde (PFA) solution was used to fix the cell spheroids for 15 min followed by a wash with PBS. Later, the tumor spheroids were stained by DAPI and were observed under CLSM.

**Animal tumor model**. The animal experiment procedures were conducted following an approved protocol (UMARE-030-2018) by the University of Macau Animal Ethics Committee.

For the construction of the subcutaneous B16F10 tumor model, $1 \times 10^6$ B16F10 cells were suspended in cold PBS and subcutaneously injected into the right flank of C57BL/6 female mice (6–8 weeks of age) to establish tumors. After 7 days' cell inoculation, tumor volumes reached ~ 100 mm$^3$, and mice are divided into different groups for further experiments.

For the construction of B16F10 tumor metastasis model, $1 \times 10^5$ B16F10-FLuc cells were implanted in each C57BL/6 female mouse (6–8 weeks of age) intravenously. 6 days later, melanoma metastasis model in the lung was established.

**In vivo fluorescence imaging**. For the in vivo fluorescence imaging experiment, when the subcutaneous B16F10 tumor volume reached around 100 mm$^3$, HGF-ICG was intravenously injected into the mice. IVIS Lumina XR was used to scan the whole mouse body at time points (2 h, 6 h, 12 h, 24 h, and 48 h). The data was calculated by the region of interest (ROI).

**Biodistribution of HGF NPs by ICP-MS**. When the subcutaneous B16F10 tumor volume was about 100 mm$^3$. Mice were randomly divided into 8 groups (0.5 h, 2 h, 4 h, 6 h, 8 h, 12 h, 24 h, 48 h). Mice were injected with HGF NPs (5 mg kg$^{-1}$ Fe$^{3+}$) intravenously. Blood, heart, liver, spleen, kidney, lung, tumor, brain, and muscle were collected and weighed. All the organs and blood were digested with 0.4 ml of H$_2$O$_2$ and 1.6 ml 63 wt % nitric acid for 5 days at room temperature. The solutions were filtered and constant the volume in 10 ml with 2 wt % nitric acid. Fe concentration in organs (2 h, 6 h, 12 h, 24 h, 48 h) and blood (0.5 h, 4 h, 8 h, 12 h, 24 h, 48 h) was determined by ICP-MS and analyzed for biodistribution and blood circulation time, respectively.

**Exosome isolation**. To isolate and enrich exosomes from cell culture supernatants, B16F10 cells were cultured in media supplemented with 10% exosome depleted FBS after diverse treatments (PBS, HACA-Fe, GW4869, or HGF NPs (GW4869, 20 μM)). Cell-free culture supernatants were then kept by centrifugation at 2,000 g for 20 min, facilitating a collection of secreted exosomes via a total exosome isolation reagent (Thermo Fisher, Cat No. 4478359). In detail, co-culturing the working reagent and cell-free supernatant overnight culturing at 2 °C to 8 °C precipitated the exosomes out. The precipitated exosomes were then collected by centrifugation at 10,000 g for 60 min (Thermo Scientific, Sorvall RC 6 Plus Centrifuge), for western blot analysis and further research.

To isolate exosome from melanoma tissues, a standard centrifugation protocol was applied[31]. Collected tumor tissues were firstly digested by slicing the samples into small fragments (1–2 mm) and incubating in RPMI-1640 medium containing collagenase IV, hyaluronidase, and deoxyribonuclease I at 37 °C for 30 min. After a filtration step to obtain single cell suspension (70 μm), cells and cellular debris were further removed by centrifugations at 300 g for 10 min and 2,000 g for 20 min, separately, to collect cell-free supernatant. Until here, there were microvesicles and exosomes we need mixing in the supernatant, and the former was discarded after centrifugation at 16,500 g for 45 min at 4 °C (Thermo Scientific, Sorvall RC 6 Plus Centrifuge). Supernatants were then ultracentrifuged at 100,000 g for 2 h at 4 °C (Beckman Coulter, Optima XPN-90) to enrich the pelleted exosomes for western blot analysis.

**Western blot analysis**. Enriched exosomes isolated from B16F10 cell supernatant or tumor tissues after diverse treatments (PBS, HACA-Fe, GW4869, HACA-GW or HGF NPs (GW4869, 20 μM)) were separated using 10% SDS–PAGE and transferred onto nitrocellulose membranes. The blots were blocked with 5 wt % non-fat dry milk at room temperature for 1 h and incubated with the relevant primary antibodies overnight at 4 °C, followed by incubation with HRP-conjugated secondary antibodies at room temperature for 1 h. The blots of CD63 and PD-L1 were detected by ECL detection reagents (Thermo Fisher, Cat No. 32209) and captured using ChemiDoc Imaging System (Bio-Rad). Anti-CD63 (Biolegend, Cat No. 143902, dilution ratio 1:2000) was used as an exosome marker. GAPDH (Santa Cruz Biotechnology, Cat No. sc-365062, dilution ratio 1:1000) was used as a loading control.

**IFN-γ release by T cell after co-culture with B16F10 cell-derived exosome**. Mouse CD8$^+$ T cells were isolated from the spleen using EasySep Mouse CD8$^+$ T Cell Isolation Kit (Stemcell, Cat No. 19853). Then cells were stimulated in Roswell Park Memorial Institute (RPMI) 1640 Medium containing anti-CD3 (2 μg ml$^{-1}$, eBioscience, Cat No. 11-0031-82) and anti-CD28 (2 μg ml$^{-1}$, eBioscience, Cat No. 16-0281-82) antibodies for 24 h. Meanwhile, B16F10 cells (20,000 cells per well) were seeded in 6-well plates and incubated with PBS, HACA-Fe, GW4869, HACA-GW, or HGF NPs (GW4869, 20 μM) for 24 h, respectively. Total exosome isolation reagent (Thermo Fisher, Cat No. 4478359) was applied to collect the secreted exosomes. In detail, the reagent was added to the cell supernatant for overnight culturing at 2 °C to 8 °C to precipitate the exosomes out. The precipitated exosomes were then achieved by standard centrifugation at 10,000 g for 60 min. Co-culturing the collected exosomes and CD8$^+$ T cells can detect the exosomal effect on T cells. After 48 h, the supernatant of CD8$^+$ T cells were collected by centrifugation at

3,000 g for 10 min and filtered at 0.2 μm. At last, the IFN-γ level was examined by ELISA kit (LifeSpan BioSciences).

**Lipid peroxidation assessed by BODIPY-C11 staining**. B16F10 cancer cells (20,000 cells per well) were seeded in 6-well plates. On the day of the experiment, cancer cells were treated with the $CD8^+$ T-cell supernatant as described above. Tumor cells were then collected by trypsinization for staining. Then cells were resuspended in 1 ml of PBS containing 5 μM of BODIPY 581/591 C11 and incubated for 20 min at 37 °C. Cells were washed for 3 times and resuspended in 300 μl of fresh PBS, then analyzed immediately with a flow cytometer (C6, BD Biosciences).

To quantify the lipid peroxidation in tumor tissue from subcutaneous B16F10 tumor model that received PBS, HACA-Fe, HACA-GW, or HGF NPs treatment, a single cell suspension was firstly prepared. Tumor tissue was collected and cut into small pieces, then mechanically minced against a 70 μm size of cell strainer, and washed with FACS buffer (1 wt % bovine serum albumin in PBS). The cell mixture was collected. The cell pellet was stained with an anti-CD45 antibody, followed by BODIPY 581/591 C11. Cells were strained through a 40 μm cell strainer and analyzed immediately with a flow cytometer. The signals from both non-oxidized C11 (PE channel) and oxidized C11 (FITC channel) were monitored. The ratio of MFI of FITC to MFI of PE was calculated for each sample. The data were normalized to control samples as shown by the relative lipid ROS.

**Expression of SLC3A2 and SLC7A11 evaluation**. The expression level of SLC3A2 and SLC7A11 in vitro was detected by CLSM. Briefly, B16F10 cells were prepared in confocal dishes for easy detection. The cells were treated with PBS, HGF NPs, and HGF NPs with IFN-γ, separately. Then, each of these groups was divided into two parts, prior to adding the primary antibody of SLC3A2 and SLC7A11 individually for 1 h at room temperature. After removal of the medium and washing the cells three times with PBS, the second antibody with FITC labeling was added into the two cell groups. The cells were finally fixed in cold ethanol for 15 min and mounted with DAPI for CLSM observation.

**GPX4 activity assay**. Intracellular GPX4 activity was measured using a cellular glutathione peroxidase assay kit (Beyotime, Cat No. S0056). B16F10 cancer cells were seeded in a 6-well plate and cultured at 5% $CO_2$, 37 °C overnight. The cells were incubated with PBS, HACA-Fe, HACA-GW, or HGF NPs for 24 h. The cell lysates were collected and measured according to the manufacturer's instructions. A microplate reader was applied to measure the absorbance at 340 nm. The tumor tissue of GPX4 activity was measured using the same assay kit. The tumor-bearing mice were sacrificed after different treatments (PBS, HACA-Fe, HACA-GW, or HGF NPs) to collect the tumor tissues and store them at − 80 °C before measurements of GPX4 activity. The activity of GPX4 in tumor tissues was measured according to the manufacturer's instructions. A microplate reader was used to measure the absorbance at 340 nm.

**Glutathione quantification**. Thirty milligrams of B16 tumor tissue from each subcutaneous B16F10 tumor-bearing animal after different treatments were homogenized in 300 μl of RIPA buffer containing protease inhibitors. Tissue homogenates were centrifuged at 10,000 g for 10 min at 4 °C and 100 μl of supernatant was used for Glutathione (GSH) analysis with a GSH-Glo Glutathione Assay (Promega) kit following the manufacturer's instruction. In brief, 100 μl of 1 × GSH-Glo Reagent was directly added to each well and followed with 30 min incubation. Then, 100 μl of reconstituted Luciferin Detection Reagent was added and mixed. After 15 min, luminescence was measured using a Microplate Reader. A standard curve for GSH concentration was generated along with samples and used for calculation. In the meantime, cell viability in another set of wells was measured using a Cell Proliferation kit (Sigma-Aldrich) following the manufacturer's instructions. Relative cell viability was calculated in comparison with the control group (100% cell viability). GSH concentration in each group was normalized to cell viability as shown by relative contents of GSH.

**Flow cytometric analysis**. Subcutaneous B16F10 tumors were established in C57/BL6 mice as described above. When the tumor volume reached ~ 100 mm$^3$, the mice were divided into four groups and injected intravenously with PBS, HACA-Fe, HACA-GW, or HGF NPs, respectively. TDLNs were collected at the 3$^{rd}$ day's post-treatment and cut into small pieces, then the tissue was mechanically minced against a 70 μm cell strainer to obtain single-cell suspension and washed with FACS buffer. Then, the cells were stained with corresponding antibodies: CD45 (Biolegend, Cat No. 103101, dilution ratio 1:50), CD11c (Biolegend, Cat No. 117306, dilution ratio 1:50), CD80 (Biolegend, Cat No. 104705, dilution ratio 1:100), CD86 (Biolegend, Cat No. 105005, dilution ratio 1:100), CD4 (eBioscience, Cat No. 11-0041-82, dilution ratio 1:50), CD8 (eBioscience, Cat No. 45-0081-82, dilution ratio 1:50), Tim3 (Biolegend, Cat No. 119718, dilution ratio 1:100), Ki67 (Biolegend, Cat No. 652411, dilution ratio 1:100) and Granzyme B (Biolegend, Cat No. 652411, dilution ratio 1:50) following the manufacturer's instructions. For the intracellular staining, after surface staining, the cell suspension was fixed and permeabilized with the commercial buffer Flow Cytometry Permeabilization/Wash

Buffer I (R&D Systems), then stained with IFN-γ antibody (R&D Systems, Cat. No. IC485P-100, dilution ratio 1:50) and analyzed by flow cytometry.

On day 14 post-treatment, tumors and spleens were collected and incubated in dissociation buffer with 1640 medium (contained collagenase IV, hyaluronidase, and deoxyribonuclease I) at 37 °C for digesting tissue. Then the tissue was mechanically minced against a 70 μm cell strainer and washed with FACS buffer. Then the cells were stained with surface antibodies: CD3 (Biolegend, Cat No. 100307, dilution ratio 1:50), CD44 (eBioscience, Cat No. 12-0441-81, dilution ratio 1:100) or CD62L-APC (eBioscience, Cat No. 17-0621-81, dilution ratio 1:50) following the manufacturer's instructions. The stained cells were detected using flow cytometry. The data were analyzed using FlowJo 10.0.

**Enzyme-linked immunosorbent assay (ELISA) for cytokine profile analysise**. The expression levels of cytokines (IFN-γ, TNF-α, IL-6 and IL-12p70) in the sera of mice were detected by commercial ELISA kits (Neobioscience Biotechnology, Shenzhen, China) by following the manufacturer's instructions.

**Immunofluorescence staining**. Tumor or TDLN tissues were cut into slices with 10 μm thickness by using Cryostat (Leica CM5030 Cryostat). Incubating these tissue slices with corresponding antibodies for 30 min at 4 °C, immunofluorescent samples can be collected with an ultimate DAPI staining.

**In vivo anti-tumor activity**. For the subcutaneous B16F10 tumor model, therapies were conducted when the tumor volume reached ~ 100 mm$^3$. All mice were divided into four groups: PBS, HACA-Fe, HACA-GW, and HGF (4.5 mg kg$^{-1}$ GW4869, 5 mg kg$^{-1}$ Fe). On day 0, 2, and 4, the mice were injected intravenously with indicated formulations. For PD-L1 blockade, an anti-mouse PD-L1 monoclonal antibody (5 mg kg$^{-1}$) was administered intraperitoneally on day 1, 3, and 5. The tumor size and body weight were measured every 3 days. The tumor volume (V) was calculated according to the following formula: width$^2$ × length/2.

In the B16F10-luc tumor metastasis model, mice were divided into four groups: PBS, anti-PD-L1, HGF, and HGF + anti-PD-L1. The dose of anti-PD-L1 antibody was 5 mg kg$^{-1}$ for each shot. As indicated in the schematic treatment procedure, HGF NPs were injected on day 6, 8, and 10, and the anti-PD-L1 antibody was administered intraperitoneally on day 7, 9, and 11. The tumor bioluminescence imaging was observed on day 12, 15, 18, and 21 by in vivo extreme system (Bruker). On day 21, the representative lungs of each group of mice were collected for imaging and analysis.

**Statistical analysis**. All quantitative data are expressed as mean ± standard deviation (SD) unless otherwise indicated. For multiple comparisons, one-way analysis of variance (ANOVA) with Tukey's post hoc test was used. Statistical analysis was performed using GraphPad Prism 8.0. P-values of < 0.05 were considered significant. *$P < 0.05$, **$P < 0.01$, ***$P\ 0.001$, and ****$P < 0.0001$.

**Reporting Summary**. Further information on research design is available in the Nature Research Reporting Summary linked to this article.

## Data availability

The pertinent data supporting the findings of this study are available within the Article, Supplementary Information, or Source Data file. Source data are provided with this paper and are also available from Figshare with the identifier https://figshare.com/s/609c457b4a3698a2b086. Source data are provided with this paper.

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

## Acknowledgements

This work was supported by the Faculty of Health Sciences, University of Macau, and the Shenzhen Science and Technology Innovation Commission, Shenzhen-Hong Kong-Macau Science and Technology Plan C (File no. SGDX20201103093600004), and the Start-up Research Grant (SRG) of the University of Macau (File no. SRG2018-00130-FHS), and the Science and Technology Development Fund, Macau SAR (File no. 0109/2018/A3 and 0011/2019/AKP). We appreciate the assistance and support from the Proteomics, Metabolomics, and Drug Development Core, Animal Research Core, and Biological Imaging and Stem Cell Core in the Faculty of Health Sciences, University of Macau.

## Author contributions

G.W. and L.X. designed and performed the majority of experiments. B.L. and G.W. analyzed and interpreted the data, and prepared the manuscript. W.S. and J.Y. assisted the in vivo therapeutic tests and immunofluorescence analysis. J.L. and H.T. contributed the nanoparticle synthesis and dye labeling. W.L. and Z.Z. assisted with the in vitro cellular experiments. Y.T. assisted with the exploration and verification of ferroptosis in the cell level. Y.D. supervised all of the experiments and revised the final manuscript.

## Competing interests

The authors declare no competing interests.
