## [Peer Review File · Nature Communications]

Reviewers' Comments:

Reviewer #1:

Remarks to the Author:

In the manuscript by Guohao Wang et al., the authors showed a novel combination therapy targeting both exosome secretion by cancer cells and ferroptosis. In their article, the authors developed a nanounit strategy that reverses immune suppression of exosomal PD-L1 and is associated with enhanced ferroptosis in vivo. This strategy is exciting, and it can increase immune response in therapy, especially when combined with PDL1 checkpoint blockade. Thus, this work is very interesting for translational research as well. This work is novel and original and may lead to improved anti-cancer therapy and unravel the role of ferroptosis in cancer therapy. The data support the main conclusions. However, there are several important issues which the authors should address before this article can be published in Nature Communications.

Major comments

1. The authors should provide clear evidence of cell death induction by ferroptosis in their in vivo models in the tumors as a result of their therapy. The authors measured SLC7A11, SLC3A2, GSH, lipid peroxidation and Cytine but did not provide direct evidence and experimental proof of cell death induction. So, a direct experimental proof (e.g., either by TUNEL staining in combination with the markers mentioned above or using other methods) should be provided of the occurrence of ferroptotic cell death in vivo under the investigated experimental conditions.
2. Next, it is essential to exclude also the contribution of other cell death modalities, for example like apoptosis and necroptosis, which can be done either using IHC (active/cleaved caspase-3 staining and RIPK3P) or by inclusion in the study additional specific cell death inhibitors such as zVAD-fmk for apoptosis and Nec-1s for necroptosis.

Minor comments

- 1) Please check English translations everywhere. Some parts are not clearly written, and spelling errors are present. Please carefully proof-read the article.
- 2) Recently, the immunogenicity of ferroptosis has been recently described in the field of immunogenic cell death (PMID: 33188036, PMID: 30686534). The authors should provide a more complete up-to-date state of the art and reflect in their discussion in regard to already published data.
- 3) CD44 function is never explained, this is not clear for the reader why cells with CD44 overexpression are targeted.
- 4) 'Quantitative plunging of tumoral exosomes': this is not clear; please explain.
- 5) In the methods section, there is no explanation about cell culture. Please add.
- 6) Multicellular spheroids means that it consists out of multiple cell types. This is not the case here.
- 7) Figure 2f: no control for GW only is added to the graph.
- 8) Figure 4o: this figure is a repetition of figure 1b.
- 9) In figure 5f other populations can be seen in the gating. This is confusing, are those cells also seen as CD44+. Also, in this figure, you mention looking at CD44^{high}, but only with high fluorescence intensity; you can say it is CD44^{high}, not the whole range. Not everything can be high.

Reviewer #2:

Remarks to the Author:

In this paper, the authors report on nano-assembly of exosome inhibitor (GW4869) and ferroptosis inducer (Fe³⁺) in amphiphilic hyaluronic acid to evoke potent anti-tumor immune response to hypermetastatic B16F10 melanoma. The results showed that combination of GW4869 and ferroptosis can significantly reduce exosomes and improve the immunotherapy for melanoma. The concept of this paper is interesting, and the paper is well-written. It can be accepted after

following revisions.

1. Fig.3a: HGF induced obviously more effective exosome inhibition than GW4869. In comparison, downregulation of PD-L1 was not so great. The claimed effect of HGF on exosomal PD-L1 can not be supported by these data. The author needs to clarify this or change their claims.
2. The authors need to explain why HGF exhibited similar reduction of exosomes to HACA-GW. Given the fact that ferroptosis would cause cell death and accordingly reduction of exosomes, it is important to have a control with HACA-Fe on the exosomes. Furthermore, it would be more important to measure the exosomal PD-L1 in the tumor tissue.
3. Are there any connections between exosomes and PD-L1 expression on the tumor cells? It would be better if the authors could collect and analyze the PD-L1 expression on both B16-F10 cells and exosomes.
4. Since there were more memory T cells in spleen, it would be more interesting to see whether there is long-term tumor inhibitory effect. In general, the therapeutic experiments are too short.

Reviewer #3:

Remarks to the Author:

The authors developed nanoparticles (HGF) by combining ceramide inhibition drug GW4869 and ferroptosis inducer Fe³⁺. By suppressing T cell function via exosomal PD-L1 and Fe³⁺-induced ferroptosis in tumor cells, HGF was claimed to inhibit tumor growth and metastasis, enhance checkpoint blockade. Using amphiphilic HACA to encapsule the hydrophobic GW4869, and targeting CD44^{high} tumor cells, the nanoparticles showed anti-tumor effect and induced T cells activation in vivo. However, there are also some a number questions in this manuscript:

1. Fig.3c, it would be hard to understand why there was no CD63 from enriched exosomes in the western blot figure. Also, how ferroptosis affects the level of PD-L1 should be addressed or at least discussed.
2. Fig.4a,b, the exosomes should be used instead of using the B16F10-cultural medium to study the effect of exosomes in this process. The media contain multiple factors and the drugs left in the medium might affect the results. Furthermore, it would be important to show the direct effect of these drugs on T cells.
3. Fig 4o, please provide direct evidence on the deduction of GPX4 in the figure, and it would be better to show Gpx4 level after applying the nanoparticles in both tumor cells and T cells.
4. Fig. 4e and Fig 4K, these two results seem to contradict. Please confirm the effect from HGF+IFN- γ group in Fig 4K on the level of SLC7A11 and SLC3A2.
5. Fig. 5F, need to provide the gating rationale or control for the CD44 high population.
6. As HA on the nanoparticle can help to target CD44 overexpressed tumor cells, how about the side effect on activate T cells also with high level of CD44?
7. GW4869 influences a number of cellular activity related to the ceramide pathway on tumor cells, not just exosome secretion.
8. There is no direct evidence that the observed inhibitory effect is from PD-L1 on the exosomes. The statement on the role of exosomal PD-L1 in the title and the text is not warranted.

REVIEWER COMMENTS

Reviewer #1 (Remarks to the Author):

In the manuscript by Guohao Wang et al., the authors showed a novel combination therapy targeting both exosome secretion by cancer cells and ferroptosis. In their article, the authors developed a nanounit strategy that reverses immune suppression of exosomal PD-L1 and is associated with enhanced ferroptosis in vivo. This strategy is exciting, and it can increase immune response in therapy, especially when combined with PDL1 checkpoint blockade. Thus, this work is very interesting for translational research as well. This work is novel and original and may lead to improved anti-cancer therapy and unravel the role of ferroptosis in cancer therapy. The data support the main conclusions. However, there are several important issues which the authors should address before this article can be published in Nature Communications.

Major comments

1. The authors should provide clear evidence of cell death induction by ferroptosis in their in vivo models in the tumors as a result of their therapy. The authors measured SLC7A11, SLC3A2, GSH, lipid peroxidation and Cytine but did not provide direct evidence and experimental proof of cell death induction. So, a direct experimental proof (e.g., either by TUNEL staining in combination with the markers mentioned above or using other methods) should be provided of the occurrence of ferroptotic cell death in vivo under the investigated experimental conditions.

2. Next, it is essential to exclude also the contribution of other cell death modalities, for example like apoptosis and necroptosis, which can be done either using IHC (active/cleaved caspase-3 staining and RIPK3P) or by inclusion in the study additional specific cell death inhibitors such as zVAD-fmk for apoptosis and Nec-1s for necroptosis.

Minor comments

- 1) Please check English translations everywhere. Some parts are not clearly written, and spelling errors are present. Please carefully proof-read the article.
- 2) Recently, the immunogenicity of ferroptosis has been recently described in the field of immunogenic cell death (PMID: 33188036, PMID: 30686534). The authors should provide a more complete up-to-date state of the art and reflect in their discussion in regard to already published data.
- 3) CD44 function is never explained, this is not clear for the reader why cells with CD44 overexpression are targeted.
- 4) 'Quantitative plunging of tumoral exosomes': this is not clear; please explain.
- 5) In the methods section, there is no explanation about cell culture. Please

add.

6) Multicellular spheroids means that it consists out of multiple cell types. This is not the case here.

7) Figure 2f: no control for GW only is added to the graph.

8) Figure 4c: this figure is a repetition of figure 1b.

9) In figure 5f other populations can be seen in the gating. This is confusing, are those cells also seen as CD44+. Also, in this figure, you mention looking at CD44^{high}, but only with high fluorescence intensity; you can say it is CD44^{high}, not the whole range. Not everything can be high.

Reviewer #2 (Remarks to the Author):

In this paper, the authors report on nano-assembly of exosome inhibitor (GW4869) and ferroptosis inducer (Fe³⁺) in amphiphilic hyaluronic acid to evoke potent anti-tumor immune response to hypermetastatic B16F10 melanoma. The results showed that combination of GW4869 and ferroptosis can significantly reduce exosomes and improve the immunotherapy for melanoma. The concept of this paper is interesting, and the paper is well-written. It can be accepted after following revisions.

1. Fig.3a: HGF induced obviously more effective exosome inhibition than GW4869. In comparison, downregulation of PD-L1 was not so great. The claimed effect of HGF on exosomal PD-L1 can not be supported by these data. The author needs to clarify this or change their claims.

2. The authors need to explain why HGF exhibited similar reduction of exosomes to HACA-GW. Given the fact that ferroptosis would cause cell death and accordingly reduction of exosomes, it is important to have a control with HACA-Fe on the exosomes. Furthermore, it would be more important to measure the exosomal PD-L1 in the tumor tissue.

3. Are there any connections between exosomes and PD-L1 expression on the tumor cells? It would be better if the authors could collect and analyze the PD-L1 expression on both B16-F10 cells and exosomes.

4. Since there were more memory T cells in spleen, it would be more interesting to see whether there is long-term tumor inhibitory effect. In general, the therapeutic experiments are too short.

Reviewer #3 (Remarks to the Author):

The authors developed nanoparticles (HGF) by combining ceramide inhibition drug GW4869 and ferroptosis inducer Fe³⁺. By suppressing T cell function via

exosomal PD-L1 and Fe³⁺-induced ferroptosis in tumor cells, HGF was claimed to inhibit tumor growth and metastasis, enhance checkpoint blockade. Using amphiphilic HACA to encapsulate the hydrophobic GW4869, and targeting CD44^{high} tumor cells, the nanoparticles showed anti-tumor effect and induced T cells activation in vivo. However, there are also some a number questions in this manuscript:

1. Fig.3c, it would be hard to understand why there was no CD63 from enriched exosomes in the western blot figure. Also, how ferroptosis affects the level of PD-L1 should be addressed or at least discussed.
2. Fig.4a,b, the exosomes should be used instead of using the B16F10-cultural medium to study the effect of exosomes in this process. The media contain multiple factors and the drugs left in the medium might affect the results. Furthermore, it would be important to show the direct effect of these drugs on T cells.
3. Fig 4o, please provide direct evidence on the deduction of GPX4 in the figure, and it would be better to show Gpx4 level after applying the nanoparticles in both tumor cells and T cells.
4. Fig. 4e and Fig 4K, these two results seem to contradict. Please confirm the effect from HGF+IFN- γ group in Fig 4K on the level of SLC7A11 and SLC3A2.
5. Fig. 5F, need to provide the gating rationale or control for the CD44 high population.
6. As HA on the nanoparticle can help to target CD44 overexpressed tumor cells, how about the side effect on activate T cells also with high level of CD44?
7. GW4869 influences a number of cellular activity related to the ceremide pathway on tumor cells, not just exosome secretion.
8. There is no direct evidence that the observed inhibitory effect is from PD-L1 on the exosomes. The statement on the role of exosomal PD-L1 in the title and the text is not warranted.

REVIEWER COMMENTS

Note to reviewers: We really appreciate the reviewers to give insightful comments to further improve our manuscript. Here, the detailed point-by-point responses have been provided to address all the comments raised by the reviewers.

(Note: corresponding changes in the manuscript are marked with red font color)

Detailed Point-by-point response to the reviewer's comments:

Reviewer #1 (Remarks to the Author): with expertise in ferroptosis and cancer immunology

In the manuscript by Guohao Wang et al., the authors showed a novel combination therapy targeting both exosome secretion by cancer cells and ferroptosis. In their article, the authors developed a nanounit strategy that reverses immune suppression of exosomal PD-L1 and is associated with enhanced ferroptosis in vivo. This strategy is exciting, and it can increase immune response in therapy, especially when combined with PDL1 checkpoint blockade. Thus, this work is very interesting for translational research as well. This work is novel and original and may lead to improved anti-cancer therapy and unravel the role of ferroptosis in cancer therapy. The data support the main conclusions. However, there are several important issues which the authors should address before this article can be published in Nature Communications.

Major comments

1. The authors should provide clear evidence of cell death induction by ferroptosis in their in vivo models in the tumors as a result of their therapy. The authors measured SLC7A11, SLC3A2, GSH, lipid peroxidation and Cytine but did not provide direct evidence and experimental proof of cell death induction. So, a direct experimental proof (e.g., either by TUNEL staining in combination with the markers mentioned above or using other methods) should be provided of the occurrence of ferroptotic cell death in vivo under the investigated

experimental conditions.

Response: The hematoxylin–eosin (H&E) staining (**Supplementary Fig. 23 in the revised supporting information**) showed obvious tumor tissue damage and the formation of many cavities, indicating that the HGF NPs (G4) induced the therapeutic death of tumor cells effectively.

As suggested, terminal deoxynucleotidyl transferase (TdT) dUTP nick-end labeling (TUNEL) method (TUNEL Andy Fluor™ 488 Detection Kit) was further applied to detect the dead cells in tumor tissue. As is shown in **Supplementary Fig. 24 below**, the brightest green fluorescent signal from dead cells can be observed in HGF NPs group (G4). Results of H&E staining and TUNEL staining concluded a robust therapeutic effect of our nanounit designed. Moreover, a detailed analysis upon the cell death modalities is shown in the ‘question #2, Reviewer #1’.

Supplementary Fig. 23. H&E-stained slice images of tumor tissue after treatment. G1, PBS; G2, HACA-Fe; G3, HACA-GW; G4, HGF. Scale bar: 200 μm .

Supplementary Fig. 24. TUNEL staining of tumor slices obtained from B16F10 tumor-bearing mice after various treatments. G1, PBS; G2, HACA-Fe; G3, HACA-GW; G4, HGF. The scale bar is 50 μm .

The supplementary Fig. 24 has been added in in the revised Supporting Information.

2. Next, it is essential to exclude also the contribution of other cell death modalities, for example like apoptosis and necroptosis, which can be done either using IHC (active/cleaved caspase-3 staining and RIPK3P) or by inclusion in the study additional specific cell death inhibitors such as zVAD-fmk for apoptosis and Nec-1s for necroptosis.

Response: To explore the modalities of cell death induced by HGF, ferroptosis inhibitor (ferrostatin-1), apoptosis inhibitor (carbobenzoxy-valyl-alanyl-aspartyl-[O-methyl]-fluoromethylketone, Z-VAD-FMK) and necroptosis inhibitor (necrostatin-1s, Nec) were adopted to co-culture with B16F10 cells, separately. From the cellular viabilities in the **Supplementary Fig. 19** below, we noticed that HGF-induced cell death was rescued by the ferroptosis inhibitor (ferrostatin-1) only, but not by the apoptosis inhibitor (Z-VAD-FMK) or the necroptosis inhibitor (Nec), which is a direct experimental proof to demonstrate the main ferroptotic cell deaths in vivo.

Furthermore, the levels of apoptosis and necroptosis in tumor tissues of different groups were detected via an immunofluorescence staining method by using apoptosis marker (cleaved caspase-3) and necroptosis marker (RIPK3P), respectively. No stronger fluorescence signal of the cleaved caspase-3 (**Supplementary Fig. 25**) or RIPK3P (**Supplementary Fig. 26**) was observed in HGF NPs group compared to the other three groups. Together, typical ferroptosis played the key role in the therapeutic function of HGF group.

Supplementary Fig. 19. HGF NPs induced ferroptotic cell death in B16F10 cells. Cell viability measurements in B16F10 cells treated with HGF and cell death inhibitors for 24 h. Ferr-1, 1 μ M ferrostatin-1 (inhibitor of ferroptosis); Z-V, 20 μ M Z-VAD-FMK (inhibitor of apoptosis); Nec, 2 μ M necrostatin-1s (inhibitor of necroptosis). Data was presented as mean \pm s.d. One-way ANOVA with Tukey's post hoc test. Significance was presented as *P < 0.05, **P < 0.01, ***P < 0.001 and ****P < 0.0001.

Supplementary Fig. 25. Cleaved caspase-3 expressions in tumor tissues were detected by immunofluorescence staining after different treatments. G1, PBS; G2, HACA-Fe; G3, HACA-GW; G4, HGF. The scale bar is 50 μm .

Supplementary Fig. 26. RIPK3P expressions in tumor tissues were detected by immunofluorescence staining after different treatments. G1, PBS; G2, HACA-Fe; G3, HACA-GW; G4, HGF. The scale bar is 50 μm .

Supplementary Fig. 19, 25 and 26 have been added into the revised Supporting Information. In addition, the corresponding context has been replenished in line 187-192 of page 7 and 220-225 of page 8-9.

Minor comments

1) Please check English translations everywhere. Some parts are not clearly written, and spelling errors are present. Please carefully proof-read the article.

Response: The articles has been proofread carefully and errors have been corrected in our revised manuscript.

For instance,

‘whether loss of exosomal PD-L1’ has been revised as ‘whether inhibiting the exosomal PD-L1’;

‘Quantitative plunging of tumoral exosomes’ has been revised as ‘Significant elimination of tumoral exosomes’;

‘Multicellular spheroids’ has been revised as ‘B16F10 cancer cellular spheroids’.

2) Recently, the immunogenicity of ferroptosis has been recently described in the field of immunogenic cell death (PMID: 33188036, PMID: 30686534). The authors should provide a more complete up-to-date state of the art and reflect in their discussion in regard to already published data.

Response: In response to your helpful suggestion, we have carefully revised the manuscript and updated the cutting-edge knowledge of ferroptosis in line 53-56 of page 2-3.

3) CD44 function is never explained, this is not clear for the reader why cells with CD44 overexpression are targeted.

Response: The cluster of differentiation protein 44 (CD44), as the main hyaluronan

binding receptor (Toole, B. P., et al. *Nat. Rev. Cancer* **2004**, 4, 528–539), overexpresses on the cancer cellular surface (Eliaz, R. E., et al. *Cancer Res.* **2001**, 61, 2592–260). However, in healthy tissues, CD44 shows much lower level and exist in intracellular area mainly. This tremendous difference of CD44 expression promotes the HA-based HGF NPs we designed to target tumor cells with overexpressed CD44 effectively and develop potent tumor-specific therapeutic performance.

The corresponding context has been added in line 71-73 of page 3, in the revised manuscript.

4) ‘Quantitative plunging of tumoral exosomes’: this is not clear; please explain.

Response: Herein, ‘quantitative plunging’ was adopted to indicate the significant elimination of tumoral exosomes, due to the functional HGF. This inappropriate vocabulary has been corrected as ‘Significant elimination of tumoral exosome’ in line 145 of page 6, in our revised manuscript.

5) In the methods section, there is no explanation about cell culture. Please add.

Response: The explanation has been added in the ‘Cell lines and culture media’ section in the ‘methods’ part of our revised manuscript.

6) Multicellular spheroids means that it consists out of multiple cell types. This is not the case here.

Response: The inappropriate vocabulary has been corrected as ‘B16F10 cancer cellular spheroids’ line 386 of page 14, in our revised manuscript.

7) Figure 2f: no control for GW only is added to the graph.

Response: As this reviewer kindly suggested, absorption of GW4869 has been

examined and added into **Fig. 2f** in our revised manuscript.

Fig. 2f The UV-Vis absorption spectra of HACA, FeCl₃, GW4869, HACA-GW and HGF NPs.

8) Figure 4o: this figure is a repetition of figure 1b.

Response: The repetitive figure 4q has been deleted in our revised manuscript.

9) In figure 5f other populations can be seen in the gating. This is confusing, are those cells also seen as CD44+. Also, in this figure, you mention looking at CD44high, but only with high fluorescence intensity; you can say it is CD44high, not the whole range. Not everything can be high.

Response: Thank you very much for your comment. The details in **Fig. 5 g** and **h** have been corrected. Additionally, the corresponding context has been added in line 233-236 of page 9, in our revised manuscript.

Fig. 5 g and h Representative flow cytometric analysis of memory T cells ($CD3^+CD8^+CD44^+CD62L^-$, gated on $CD3^+CD8^+$ T cells) in the spleen. $n=5$. G1, PBS; G2, HACA-Fe; G3, HACA-GW; G4, HGF. Data was presented as mean \pm s.d. One-way ANOVA with Tukey's post hoc test. Significance was presented as * $P < 0.05$, ** $P < 0.01$, *** $P < 0.001$ and **** $P < 0.0001$.

Reviewer #2 (Remarks to the Author): with expertise in hyaluronic acid-based nanosystems

In this paper, the authors report on nano-assembly of exosome inhibitor (GW4869) and ferroptosis inducer (Fe^{3+}) in amphiphilic hyaluronic acid to evoke potent anti-tumor immuno response to hypermetastatic B16F10 melanoma. The results showed that combination of GW4869 and ferroptosis can significantly reduce exosomes and improve the immunotherapy for melanoma. The concept of this paper is interesting, and the paper is well-written. It can be accepted after following revisions.

1. Fig.3a: HGF induced obviously more effective exosome inhibition than GW4869. In comparison, downregulation of PD-L1 was not so great. The claimed effect of HGF on exosomal PD-L1 can not be supported by these data. The author needs to clarify this or change their claims.

Response: Compared with PBS and GW4869 groups, we observed that HGF had the most significant inhibition on tumoral exosomes, demonstrated by the intensity of exosome biomarker CD63. However, the gray intensity of PD-L1 of the HGF group

was higher than that of the CD63, which might be due to that the content of exosomal PD-L1 was higher than the content of CD63 on the secreted exosome. Overdose of total protein thus cannot highlight the PD-L1 variation very well.

To make the results more convincing, we further optimized the experimental protocol, via reducing the amount of protein loaded and using PD-L1 antibody with higher sensitivity, to repeat the experiment and confirm the inhibition effect of HGF on exosomal PD-L1. As anticipated, higher intensity of PD-L1 was found in each group, compared to the intensity of CD63. GW4869 from HGF NPs diminished the exosome biomarker CD63 of B16F10 cells and exosomal PD-L1 dramatically. **Fig. 3a** has been corrected in our revised manuscript.

Fig. 3a Western blot analysis for exosome marker CD63 and PD-L1 in the medium of B16F10 cells after treatment. GAPDH served as a loading control. Images were representative of three experiments.

2. The authors need to explain why HGF exhibited similar reduction of exosomes to HACA-GW. Given the fact that ferroptosis would cause cell death and accordingly reduction of exosomes, it is important to have a control with HACA-Fe on the exosomes. Furthermore, it would be more important to measure the exosomal PD-L1 in the tumor tissue.

Response: From our previous western blot figure, CD63 intensities in HACA-GW and HGF groups may look similar by our naked eyes. However, obvious quantitative

differences can be achieved by using the Image J software, as shown in the Explanatory figure 1.

Explanatory figure 1. CD63 protein level of different groups.

Additionally, overdose of total protein in the western blot trial may cloud CD63 differences in HACA-GW and HGF groups further. Therefore, in our new trail with an added HACA-Fe group, loading amount of total protein was optimized (**Fig. 3c** in the revised manuscript). Lowest level of CD63 detected in HGF group convinced the effective exosome reduction via the combination of GW4869 and enhanced ferroptosis. To further analyze the effect of HACA-Fe on exosome secretion in vitro (**Fig. 3a**) and in vivo (**Fig. 3c**) through the western blot assay, we noticed that the secretion of exosomes decreased slightly, which may be explained by cellular ferroptosis.

Finally, exosomal PD-L1 level in the tumor tissue was investigated (**Fig. 3c**). HACA-GW reduced the exosomal PD-L1 effectively. A combination with ferroptosis inhibited the exosomal PD-L1 in HGF group further. **Fig. 3a** and **3c** have been renewed in our revised manuscript. The corresponding context has been added in line 135-139 of page 5 and 6, line 141-144 of page 6, in the revised manuscript.

Fig. 3a Western blot analysis for exosome marker CD63 and PD-L1 in the medium of B16F10 cells after treatment. GAPDH served as a loading control. Images were representative of three experiments.

Fig. 3c Western blot analysis of exosome marker CD63 and PD-L1 in tumor tissues after treatment. GAPDH served as a loading control. Images were representative of three experiments.

3. Are there any connections between exosomes and PD-L1 expression on the tumor cells? It would be better if the authors could collect and analyze the PD-L1 expression on both B16-F10 cells and exosomes.

Response: Thanks for the reviewer's insightful question. Recent studies showed that exosomal PD-L1 has the same membrane topology as cell surface PD-L1, with its extracellular domain exposed on the surface of the exosomes. A similar PD-L1 level exists in mouse metastatic melanoma B16F10 cells and purified exosomes (Chen, G.,

et al. *Nature* **2018**, 560, 382-386). However, some cancer cells can secrete a vast majority of their PD-L1 on exosomes rather than present PD-L1 on their cell surface, such as prostate cancer cell lines (PC3) (Poggio, M., et al. *Cell* **2019**, 177, 414-427). Therefore, the expression level of exosomal PD-L1 varies in different cancer cell lines.

To explore the PD-L1 levels on B16F10 cells and their secreted exosomes in vitro and in vivo, we collected the cancer cells and the corresponding exosomes together. An immunoblot analysis confirmed that a consistent PD-L1 level on tumor cell and exosome in **Supplementary Fig. 7**.

Supplementary Fig. 7. Immunoblots for PD-L1 in the whole cell lysate and purified exosomes from mouse melanoma B16F10 cells and tissue. The same amount of protein was loaded in each lane.

Supplementary Fig. 7 has been added into the revised supporting information. And the corresponding context has been added in line 132-134 of page 5, in the revised manuscript.

4. Since there were more memory T cells in spleen, it would be more interesting to see whether there is long-term tumor inhibitory effect. In general, the therapeutic experiments are too short.

Response: Long-term tumor inhibitory effect and survival curves had been included in the **Fig. 5b, c, d and Fig. 6b, c, d** in the revised manuscript. The corresponding

context has been corrected in line 220-226 of page 8-9 and line 241-246 of page 9, in the revised manuscript.

Fig. 5 b, c Individual B16F10 tumor growth curves (b) and average tumor growth curves (c) after treatment. The experiment was performed twice with similar results. $n = 5$ biologically independent animals per group. **d** Survival curves after treatment. G1, PBS; G2, HACA-Fe; G3, HACA-GW; G4, HGF. $n=5$. Data was presented as mean \pm s.d. The analysis method was One-way ANOVA with Tukey's post hoc test. Significance was presented as * $P < 0.05$, ** $P < 0.01$, *** $P < 0.001$ and **** $P < 0.0001$.

Fig. 6 b, c Individual B16F10 tumor growth curves (b) and average tumor growth curves (c) after treatment. The experiment was performed twice with

similar results. n = 5 biologically independent animals per group. **d** Survival curves after treatment. G1, PBS; G2, anti-PD-L1; G3, HGF; G4, HGF+ anti-PD-L1. n=5. Data was presented as mean \pm s.d. The analysis method was One-way ANOVA with Tukey's post hoc test. Significance was presented as *P < 0.05, **P < 0.01, ***P < 0.001 and ****P < 0.0001.

Reviewer #3 (Remarks to the Author): with expertise in exosomal-PDL1 and cancer immunology

The authors developed nanoparticles (HGF) by combining ceramide inhibition drug GW4869 and ferroptosis inducer Fe³⁺. By suppressing T cell function via exosomal PD-L1 and Fe³⁺-induced ferroptosis in tumor cells, HGF was claimed to inhibit tumor growth and metastasis, enhance checkpoint blockade. Using amphiphilic HACA to encapsule the hydrophobic GW4869, and targeting CD44high tumor cells, the nanoparticles showed anti-tumor effect and induced T cells activation in vivo. However, there are also some a number questions in this manuscript:

1. Fig.3c, it would be hard to understand why there was no CD63 from enriched exosomes in the western blot figure. Also, how ferroptosis affects the level of PD-L1 should be addressed or at least discussed.

Response: Exosomes in **Fig. 3c** were isolated from tumor tissues with the same weight after different treatments. Since HACA-GW and HGF NPs could inhibit the exosome secretion from the tumor site significantly, low level of exosome biomarker CD63 in these two groups were therefore collected.

To analyze the effect of ferroptosis on exosomal PD-L1, four groups including PBS, HACA-Fe, HACA-GW and HGF were prepared, in which HACA-Fe could cause ferroptosis individually. Ferroptotic cell death in HACA-Fe group reduced the total secretion of exosomes sourcing from cancer cells (**Fig. 3a** and **3c**).

Fig. 3a and **3c** have been renewed in the revised manuscript, and the corresponding context has been added in line 135-139 of page 5-6, line 141-144 of page 6, in the revised manuscript.

Fig. 3a Western blot analysis for exosome marker CD63 and PD-L1 in the medium of B16F10 cells after treatment. GAPDH served as a loading control. Images were representative of three experiments.

Fig. 3c Western blot analysis of exosome marker CD63 and PD-L1 in tumor tissues after treatment. GAPDH served as a loading control. Images were representative of three experiments.

2. Fig.4a, b, the exosomes should be used instead of using the B16F10-cultural medium to study the effect of exosomes in this process. The media contain multiple factors and the drugs left in the medium might affect the results. Furthermore, it would be important to show the direct effect of these drugs on T

cells.

Response: It is true that many factors in the cell culture medium that may affect the results, and that is why the total exosome isolation reagent (Thermo Fisher, Cat. No. 4478359) was applied in Fig. 4a to collect the exosomes first. To avoid the misunderstanding, the experimental details in the method section has been perfected as: ‘Total exosome isolation reagent (Thermo Fisher, Cat No. 4478359) was applied to collect the secreted exosomes. In detail, the reagent was added to the cell supernatant for overnight culturing at 2 °C to 8 °C to precipitate the exosomes out. The precipitated exosomes were then achieved by standard centrifugation at 10,000 g for 60 min. Co-culturing the collected exosomes and CD8⁺ T cells can detect the exosomal effect on T cells.’

As this reviewer suggested, the direct effect of different drugs on T cells has been measured. **Supplementary Fig. 3a** indicated that HACA-GW had no effect on the release of IFN- γ in T cells comparing with PBS. However, level of IFN- γ decreased in HGF and HACA-Fe groups by 9.7% and 8.6%, respectively. We also tested the cell viability of T cells after incubation with PBS, HACA-Fe, HACA-GW and HGF NPs. At the highest concentration of HACA-Fe and HGF NPs, the viability of T cells was slightly inhibited and remained above 80% (81.2% at 100 μ M for 48 h) (**Supplementary Fig. 3b.** in our revised manuscript). Thus, the inhibition of IFN- γ in these two groups might be due to the decreased cell viability induced by Fe³⁺-relevant ferroptosis.

Supplementary Fig. 3. a Relative IFN- γ release from CD8⁺ T cells incubated in PBS, HACA-Fe, HACA-GW and HGF. **b** Viability of CD8⁺ T cells after incubating with HACA, HACA-Fe, HACA-GW and HGF NPs at diverse concentrations for 48 h. n=5. Data was presented as mean \pm s.d. The analysis method was One-way ANOVA with Tukey's post hoc test. Significance was presented as *P < 0.05, **P < 0.01, ***P < 0.001 and ****P < 0.0001.

Supplementary Fig. 3a and 3b have been added into the revised Supporting Information, and the corresponding context has been added in line 110-113 of page 4-5, in the revised manuscript.

3. Fig 4o, please provide direct evidence on the deduction of GPX4 in the figure, and it would be better to show Gpx4 level after applying the nanoparticles in both tumor cells and T cells.

Response: The cystine/GSH depletion correlates with the inactivation of GPX4 positively (Wan Seok Y., et al. *Cell*, **2014**, 156, 317-331). For B16F10 tumor cells, treatment of HGF NPs downregulated the GSH level remarkably, compared with PBS and HACA-Fe groups (**Fig. 4m**). A corresponding decreasing of GPX4 activity was thus triggered (**Fig. 4e, n**) via using tert-butylhydroperoxide (tBuOOH) as a substrate and monitoring the rate of NADPH oxidation. However, treating T cells with HGFs increased GPX4 activity surprisingly (**Supplementary Fig. 22**). Such phenomena

may be due to the upregulated cystine and cysteine transporter activities in activated T cells, increasing the GSH synthesis and GPX4 activity (Ishii, T., et al. *J Cell Physiol* **1987**, 133, 330-336; Garg, S. K., et al. *Antioxidants & Redox Signaling* **2010**, 15, 39-47; Levring, T. B., et al. *Sci Rep* **2012**, 2, 266). Collectively, HGF NPs upregulated the level of GPX4 activity in T cells, but downregulated whose function in tumor cells conversely, by virtue of the reduction of exosomes and the enhanced ferroptosis (**Fig. 3d-f**). Different roles in T cells and tumor cells may associate with the robust activation of CD8⁺ and CD4⁺ T in our nanounit system.

Fig. 4e GPX4 activity in B16F10 cells after different treatments. n=5. Data was presented as mean \pm s.d. The analysis method was One-way ANOVA with Tukey's post hoc test. Significance was presented as *P < 0.05, **P < 0.01, ***P < 0.001 and ****P < 0.0001.

Fig. 4n GPX4 activity in melanoma tumor tissue after different treatments. 1, PBS; G2, HACA-Fe; G3, HACA-GW; G4, HGF. n=5. Data was presented as mean \pm s.d. The analysis method was One-way ANOVA with Tukey's post hoc test. Significance was presented as *P < 0.05, **P < 0.01, ***P < 0.001 and ****P < 0.0001.

Supplementary Fig. 22. GPX4 activity in T cells collected from lymph node after different treatments. G1, PBS; G2, HACA-Fe; G3, HACA-GW; G4, HGF. n=5. Data was presented as mean \pm s.d. The analysis method was One-way ANOVA with Tukey's post hoc test. Significance was presented as *P < 0.05, **P < 0.01, ***P < 0.001 and ****P < 0.0001.

Fig. 3d-f Flow cytometric plots (d) and quantification of CD8⁺ (e) and CD4⁺ (f) T cells, respectively, gated by CD3⁺ T cells in the TDLN (n = 5) in groups of PBS (G1), HACA-Fe (G2), HACA-GW (G3), HGF (G4) and HGF+liproxstatin (G5). n=5. Data was presented as mean ± s.d. The analysis method was One-way ANOVA with Tukey's post hoc test. Significance was presented as *P < 0.05, **P < 0.01, ***P < 0.001 and ****P < 0.0001.

Fig. 4e, 4 n and Supplementary Fig. 22 have been added in the revised manuscript, and the corresponding context has been added in line 197-200, 211-212 and 215-217 of page 8, in the revised manuscript. Experimental details about GPX4 detection have been added into the methods section.

4. Fig. 4e and Fig 4K, these two results seem to contradict. Please confirm the effect from HGF+IFN- γ group in Fig 4K on the level of SLC7A11 and SLC3A2.

Response: In fact, results of **Fig. 4e** (it has been renamed as **Fig. 4d** in the revised manuscript) and **Fig. 4k** support mutually, since IFN- γ and IFN- γ inhibitor (IFN- γ antibody) were applied in these two tests, separately. In **Fig. 4e**, the expression of the two proteins was reduced after adding the IFN- γ artificially, suggesting that IFN- γ played key role in inhibiting expression of SLC7A11 and SLC3A2. To confirm this conclusion further, IFN- γ inhibitor (anti-IFN- γ antibody) was used in **Fig. 4k** prior to HGF therapy (HGF + anti-IFN- γ group). The reduction of SLC7A11 and SLC3A2 induced by HGF can be relieved. As we claimed in line 196-197 and 208-210 of page 8, these results suggested that HGF NPs dramatically downregulated SLC7A11 and SLC3A2 by increased secretion of IFN- γ .

5. Fig. 5F, need to provide the gating rationale or control for the CD44 high population.

Response: The gating rationale for the CD44⁺CD62L⁻ T cells has been provided as **Supplementary Fig. 29** in our revised manuscript.

Supplementary Fig. 29. Representative flow cytometry gating strategies for CD3⁺CD8⁺CD44⁺CD62L⁻ T cells panel in spleen.

6. As HA on the nanoparticle can help to target CD44 overexpressed tumor cells, how about the side effect on activate T cells also with high level of CD44?

Response: We tested the potential side effect of different drugs on T cells. EasySep Mouse CD8⁺ T Cell Isolation Kit (Stemcell, 19853) was applied firstly to obtain CD8⁺ T cells from resuspended cells in the spleen. After being stimulated with anti-CD3 (2 μg ml⁻¹, eBioscience, 11-0031-81) and anti-CD28 (2 μg ml⁻¹, eBioscience, 16-0281-82) antibodies for 24 h, the CD8⁺ T cells were seeded in 96-well plates at a density of 2 × 10⁴ per well overnight. After that, cells were treated with PBS, HACA-Fe, HACA-GW or HGF NPs for 48 h, respectively. Cell viability was measured using a MTT (3-[4, 5-dimethylthiazol-2-yl]-2, 5 diphenyl tetrazolium bromide) assay, as previously described (Zhang, Y. L., et al. *Cell Cycle* **2019**, 18, 773-783; Koppula, P., et al. *Journal of Biological Chemistry* **2017**, 292,

14240-14249). At the highest concentration of HGF NPs, the viability of T cells was remained above 80% (81.2 % at 100 μ M for 48 h) (**Supplementary Fig. 3b** in our revised manuscript). The result showed that these drugs have no significant obvious inhibitory effect on the viability of T cells.

Supplementary Fig. 3b. Viability of CD8⁺ T cells after incubating with HACA, HACA-Fe, HACA-GW and HGF NPs at diverse concentrations for 48 h.

Supplementary Fig. 3b has been added into the revised Supporting Information, and the corresponding context has been added in line 111-113 of page 5, in the revised manuscript.

7. GW4869 influences a number of cellular activity related to the ceramide pathway on tumor cells, not just exosome secretion.

Response: The ceramide could promote cancer cellular apoptosis and suppress its proliferation and metastasis (Zhang J., et al. *Proc Natl Acad Sci* **1996**, 93, 5325-5328 Kitatani K., et al. *Oncogene* **2016**, 35, 2801–2812). GW4869, as a ceramide inhibitor, has the potent capacity to influence the ceramide pathway, impeding the apoptosis of cancer cell. This apoptosis inhibition of GW4869 develops under reasonable concentration, given that its overdose cytotoxicity (Qian C., et al. *Acta Pharmacol Sin* **2018**, 39, 561-568).

In our study, safe dosage of GW4869 (20 $\mu\text{mol L}^{-1}$) was applied during all experiments, from the cellular cytotoxicity test in **Supplementary Fig. 2**. In this scenario, another important function of GW4869 could develop very well—inhibiting the exosomal secretion of cancer cell.

8. There is no direct evidence that the observed inhibitory effect is from PD-L1 on the exosomes. The statement on the role of exosomal PD-L1 in the title and the text is not warranted.

Response: We thank the reviewer for asking this insightful question. Actually, ‘exosomal PD-L1’ in the title and the text is stated under our strict consideration. Poggio, M., et al. revealed inhibiting exosome secretion from PC3 cancer cells could promote the release of pro-inflammatory IL-2 cytokine from Jurkat T cells (Poggio, M., et al. *Cell* **2019**, 177, 414-427). To explore further, exosomes without PD-L1 expression, engineered by CRISPR/Cas9 technique, did not affect the pro-inflammatory immune response. Both evidences ascertained the unique significance of PD-L1 from exosome on anti-tumor immune suppression. Moreover, Chen, G., et al. discovered that blocking PD-L1 on exosome recovered the proliferation, cytokine production and cytotoxicity of CD8 T cells, confirmed on several types of cancer cell lines (Chen, G., et al. *Nature* **2018**, 560, 382-386). Therefore, exosomal PD-L1 associates with anti-tumor immunity directly.

To examine the suppressive effects of exosomal PD-L1 on CD8⁺ T cells, we incubated CD8⁺ T cells with purified B16F10 cell-derived exosomes with or without anti-PD-L1 antibody blocking for 48 h (**Supplementary Fig. 17**). B16F10 exosomes intensively inhibited the IFN- γ production of CD8⁺ T cells, whereas treating exosomes with anti-PD-L1 antibodies abolished the effect. Thus, this evidence directly confirmed that exosomal PD-L1 is strongly associated with suppression of anti-tumor immunity.

Supplementary Fig. 17. IFN- γ level in CD8⁺ T cells (stimulated with anti-CD3/CD28 antibodies) after treatment with PBS (Control), B16F10 cell-derived exosomes (Exosomes), B16F10 cell-derived exosomes with blocking PD-L1 antibodies (Exosomes+anti-PD-L1). Data were presented as mean \pm s.d. One-way ANOVA with Tukey's post hoc test. Significance was presented as *P < 0.05, **P < 0.01, ***P < 0.001 and ****P < 0.0001.

Supplementary Fig. 17 has been added into the revised Supporting Information, and the corresponding context has been added in line 174-177 of page 7, in the revised manuscript.

Reviewers' Comments:

Reviewer #1:

Remarks to the Author:

The authors have addressed the majority of the comments and I think that this manuscript can be accepted in the journal.

Reviewer #2:

Remarks to the Author:

The authors have made adequate revisions according to the comments. The paper can now be accepted for publication.

Reviewer #3:

Remarks to the Author:

The authors, in general, addressed my questions.